# ReconDreamer-RL: Enhancing Reinforcement Learning via Diffusion-based Scene Reconstruction

## Abstract

Reinforcement learning for training end-to-end autonomous driving models in closed-loop simulations is gaining growing attention. However, most simulation environments differ significantly from real-world conditions, creating a substantial simulation-to-reality (sim2real) gap. To bridge this gap, some approaches utilize scene reconstruction techniques to create photorealistic environments as a simulator. While this improves realistic sensor simulation, these methods are inherently constrained by the distribution of the training data, making it difficult to render high-quality sensor data for novel trajectories or corner case scenarios. Therefore, we propose *ReconDreamer-RL*, a framework designed to integrate video diffusion priors into scene reconstruction to aid reinforcement learning, thereby enhancing end-to-end autonomous driving training. Specifically, in *ReconDreamer-RL*, we introduce ReconSimulator, which combines the video diffusion prior for appearance modeling and incorporates a kinematic model for physical modeling, thereby reconstructing driving scenarios from real-world data. This narrows the sim2real gap for closed-loop evaluation and reinforcement learning. To cover more corner-case scenarios, we introduce the Dynamic Adversary Agent (DAA), which adjusts the trajectories of surrounding vehicles relative to the ego vehicle, autonomously generating corner-case traffic scenarios (e.g., cut-in). Finally, the Cousin Trajectory Generator (CTG) is proposed to address the issue of training data distribution, which is often biased toward simple straight-line movements. Experiments show that *ReconDreamer-RL* improves end-to-end autonomous driving training, outperforming imitation learning methods with a $5\times$ reduction in the Collision Ratio.

## 1 Introduction

End-to-end training of autonomous driving models (Zhang et al., 2023; Liao et al., 2024; Li et al., 2024b; Sun et al., 2024; Weng et al., 2024; Huang et al., 2024b; Lu et al., 2025a; Zeng et al., 2025) through reinforcement learning in closed-loop simulation environments attracts growing interest. Compared to imitation learning, which relies solely on expert-collected demonstrations, closed-loop reinforcement learning enables the model to interact with the environment, enhancing its robustness and adaptability across diverse scenarios.

Despite these advantages, existing methods still face significant challenges. One key issue is creating realistic driving environments capable of effective interactions with autonomous driving policies. Game-engine-based simulators (Zhou et al., 2023; Automotive, 2023) lack sensor-level realism, whereas real-world closed-loop training is costly and risky. To overcome these limitations, recent approaches (Gao et al., 2025; Wu et al., 2023; Yang et al., 2023c; Lu et al., 2024; Ding et al., 2025a;b) employ scene reconstruction methods to build photorealistic digital twins of real-world scenarios. However, these reconstruction-based methods are constrained by their training data, typically generating high-quality sensor outputs only within regions covered by recorded camera trajectories. Additionally, these methods fail to adequately account for corner cases such as sudden braking, as such rare behaviors are typically absent from the reconstruction data.

In this paper, we introduce *ReconDreamer-RL*, a framework that integrates the video diffusion priors (Wang et al., 2023; Zhao et al., 2024b) into scene reconstruction to enhance end-to-end au-

Figure 1: In *ReconDreamer-RL*, ReconSimulator improves appearance modeling by ReconDreamer and incorporates physical modeling to reconstruct driving scenes. In the imitation learning stage, DAA generates corner-case scenario trajectories, while CTG diversifies the ego vehicle's actions and uses ReconSimulator to render sensor data for training the policy. In the reinforcement learning stage, the policy is trained in a closed-loop environment, interacting with DAA-controlled vehicles.

tonomous driving training by reinforcement learning. The framework consists of three components: the ReconSimulator, the Dynamic Adversary Agent (DAA), and the Cousin Trajectory Generator (CTG). *ReconDreamer-RL* enhances the training process in two key stages: (1) The imitation learning stage uses behavior cloning for plan initialization. (2) The reinforcement learning stage optimizes the policy through closed-loop trial-and-error interactions with the environment. Specifically, ReconSimulator employs 3D Gaussian Splatting (3DGS) to reconstruct the driving scene and incorporates the video diffusion prior for appearance modeling. Meanwhile, a kinematic model is used to ensure the validity of vehicles' trajectories during ego-vehicle movement and scene editing. Then, in the first stage, we propose that DAA enrich corner case scenarios by controlling surrounding vehicles to generate challenging situations such as sudden lane changes. Meanwhile, the CTG enhances the diversity of sensor-collected data by synthesizing new trajectories from expert trajectories, addressing the data bias toward straight-line driving. In the second stage, the autonomous driving policy is trained within ReconSimulator via reinforcement learning, while the DAA continues to dynamically alter surrounding vehicle trajectories to create corner cases.

Through evaluation in the closed-loop 3DGS environment, we demonstrate that the end-to-end driving policy trained with the *ReconDreamer-RL* framework performs better, especially in corner cases, with a 5× reduction in Collision Ratio compared to imitation learning methods.

The main contributions of this paper are summarized as follows:

- We introduce *ReconDreamer-RL*, the framework that enhances end-to-end autonomous driving training by scene reconstruction with diffusion prior. It uses ReconSimulator to create realistic and freely explorable environments for reinforcement learning by enhancing appearance and physical modeling, reducing the sim2real gap.

- We propose DAA to auto-generate corner case instances to improve scenario coverage. Additionally, the CTG is introduced to address data distribution challenges by enriching sensor-collected expert trajectories, ensuring more balanced and comprehensive training data. This joint design further promotes model generalization and resilience.

- We validate the effectiveness of *ReconDreamer-RL* in a closed-loop 3DGS environment. *ReconDreamer-RL* performs stronger in challenging closed-loop evaluations, achieving a 5× reduction in collision rate compared to imitation learning methods.

## 2 RELATED WORK

### 2.1 DRIVING SCENE RECONSTRUCTION WITH DIFFUSION PRIOR

3DGS has become a prominent technique in 3D scene reconstruction. Various works (Yan et al., 2024; Chen et al., 2023; Huang et al., 2024a; Chen et al., 2024b; Li et al., 2025) have applied 3DGS to autonomous driving scenarios. However, these methods exhibit significant performance degradation when rendering trajectories that deviate from the training distribution. Although some methods (Yan et al., 2024; Zhou et al., 2024) leverage 3DGS with decomposed foreground and background layers to ensure accurate foreground localization, the background still suffers from sub-

stantial artifacts under novel viewpoints, for example, blurred lane markings. This poses a challenge for end-to-end autonomous driving policies, which rely heavily on background cues such as lane boundaries for reliable decision-making, making these methods unsuitable for closed-loop evaluation or as reinforcement learning simulators. To address this issue, some studies leverage video diffusion models (Rombach et al., 2022; Blattmann et al., 2023) to enhance the representation of driving scenes. DriveDreamer4D (Zhao et al., 2024a) is the first to leverage the video diffusion prior to generate novel trajectory videos for training reconstruction models. ReconDreamer (Ni et al., 2024) further improves reconstruction by integrating the video diffusion model with online artifact correction, enabling better rendering of large maneuvers. ReconDreamer++ (Zhao et al., 2025) enhances rendering quality by reducing domain gaps and refining the ground surface. However, these works focus on improving scene representation. In contrast, we leverage the video diffusion prior to improve the simulator's appearance modeling and introduce physical modeling, enhancing end-to-end autonomous driving training by reinforcement learning.

In terms of scene generation methods, DriveDreamer-2 (Zhao et al., 2024b) allows users to create corner case scenarios more effectively. However, its generation method is relatively inefficient and unsuitable for reinforcement learning training. In contrast, DiffScene (Xu et al., 2025) introduces a diffusion-based framework that efficiently generates high-quality, safety-critical driving scenarios, significantly enhancing the evaluation of autonomous vehicles.

## 2.2 END-TO-END AUTONOMOUS DRIVING

Recent advancements in end-to-end autonomous driving algorithms showcase the immense potential of learning-based planning, where raw sensor inputs are directly mapped to control outputs. UniAD (Hu et al., 2023) integrates various perception tasks (Delavari et al., 2025; Di Palo & Johns, 2024; Cheng et al., 2024; Yan et al., 2025; Zhu et al., 2022; Lu et al., 2025b; Zhang et al., 2024) to enhance planning performance. VAD (Jiang et al., 2023) focuses on improving trajectory planning by utilizing compact vectorized scene representations. Furthermore, VADv2 (Chen et al., 2024a) extends this by introducing a framework that models probability distributions over planning vocabularies. However, methods based on imitation learning rely heavily on expert demonstrations and suffer from poor generalization. To address this issue, some approaches (Li et al., 2024a; Zhang et al., 2025; Caesar et al., 2021; Shalev-Shwartz et al., 2016; Gao et al., 2024; Jiang et al., 2025) utilize deep reinforcement learning (Schrittwieser et al., 2020; Silver et al., 2016; Jumper et al., 2021; Mnih et al., 2013; Schulman et al., 2017; Ouyang et al., 2025; Shao et al., 2024) for training end-to-end autonomous driving models. Nevertheless, these methods often encounter limitations, either due to simulation environments lacking photorealistic fidelity (Dosovitskiy et al., 2017) or reliance on open-loop data. Recent work, RAD (Gao et al., 2025), introduces a reinforcement learning framework specifically designed for training autonomous driving agents in photorealistic 3DGS environments. However, RAD (Gao et al., 2025) still suffers from inadequate supervision for policy learning due to limitations in rendering novel views and a lack of corner-case scenarios in reconstructed data, leading to limited generalization and a persistent sim-to-real gap.

## 3 METHOD

### 3.1 PRELIMINARY

ReconDreamer (Ni et al., 2024) integrates the video diffusion prior to improve the performance of reconstruction methods in handling large maneuvers. The core of ReconDreamer is DriveRestorer, designed to restore artifacts in rendered videos through an online restoration process, which mitigates distortions and enhances consistency across frames. It is fine-tuned based on the video diffusion models (Wang et al., 2023; Zhao et al., 2024b) and uses a diffusion loss as follows:

$$\mathcal{L}_{\mathcal{R}} = \mathbb{E}_{\boldsymbol{z}, \epsilon \sim \mathcal{N}(0,1), t} \left[ \|\epsilon_t - \epsilon_\theta \left( \boldsymbol{z}_t, t, \boldsymbol{c} \right)\|_2^2 \right], \tag{1}$$

where $\epsilon_t$ is the random noise at time step $t$, $\epsilon_\theta$ is the denoising network used to predict the noise, $zt$ is the noisy latent variable at step $t$, and $\boldsymbol{c}$ represents the control conditions, which include the degraded video $\hat{V}$novel, 3D bounding boxes, and HDMaps, thus incorporating both low-level visual cues and high-level structural information to guide accurate video restoration.

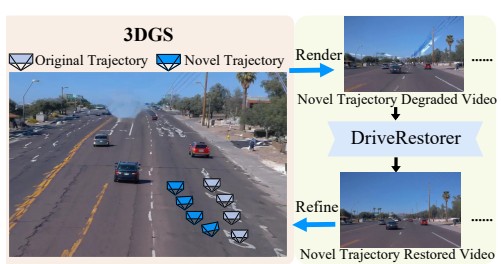 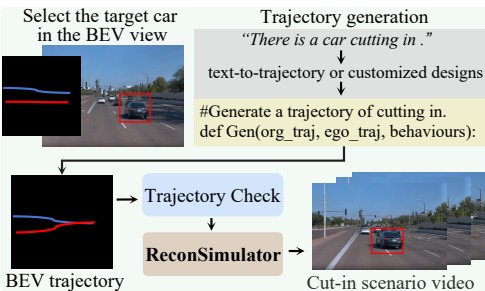

Figure 2: The process of integrating the diffusion prior for appearance modeling. During the reconstruction of driving scenes, we first render novel trajectory view videos. These rendered videos are then processed by the DriveRestorer to enhance their visual quality, and the restored results are used to further optimize the reconstruction model. This iterative process continues until the reconstruction model converges.

Figure 3: The pipeline of the DAA. DAA identifies the target vehicles based on their distances to the ego car from the BEV view, where the blue line represents the ego cars trajectory and the red line represents the target vehicle. Then, DAA generates novel trajectories based on the specified interactive behavior. The generated trajectories are checked, and feasible ones are rendered using ReconSimulator.

During the inference stage, DriveRestorer freezes its network parameters and restores novel trajectory renderings. The inference process is expressed as:

$$V_{\text{novel}} = \mathcal{R}(\hat{V}_{\text{novel}}, s), \tag{2}$$

where $\mathcal{R}$ represents the DriveRestorer model, and $s$ denotes the structural conditions corresponding to the degraded video $\hat{V}_{\text{novel}}$ (such as 3D bounding boxes and HDMaps). In *ReconDreamer-RL*, we leverage DriveRestorer to enhance scene rendering quality, enabling free ego car navigation and scene editing, as detailed in the ReconSimulator Section.

## 3.2 OVERVIEW OF THE *ReconDreamer-RL* FRAMEWORK.

Existing end-to-end autonomous driving algorithms are typically trained based on imitation learning, leading to poor generalization. To address this, RAD (Gao et al., 2025) trains the policy in 3DGS-based simulations, which provide photorealistic sensor data. However, it still faces challenges in rendering novel views and covering diverse corner cases.

We introduce the *ReconDreamer-RL* pipeline in Fig. 1. ReconSimulator first reconstructs the driving scene, providing high-fidelity sensor data for ego vehicle navigation by integrating video diffusion priors and physical modeling, ensuring more accurate and realistic environmental representations. In the imitation learning stage, DAA generates corner cases specifically designed for imitation learning, while CTG improves driving behavior through a variety of diverse and challenging actions. In the reinforcement learning stage, the policy is trained in closed-loop environments, with DAA generating new corner case trajectories to increase difficulty and enhance the robustness of the policy.

## 3.3 RECONSIMULATOR

The ideal reinforcement learning environment for autonomous driving should minimize the sim2real gap. However, existing 3DGS-based methods merely reconstruct data, failing to ensure realistic appearances along novel trajectories. Meanwhile, it is essential to ensure that the trajectories of all vehicles adhere to physical constraints. Therefore, we propose ReconSimulator to address appearance and physical modeling challenges. In terms of appearance, we integrate the diffusion prior to enhance scene representation for high-quality sensor data rendering on novel trajectories. For physical modeling, kinematic modeling is used to ensure the physical feasibility of vehicle trajectories. Next, we delve into the details of the appearance and physical modeling.

**Appearance Modeling.** To ensure high-quality rendering and scene editing, we first use 3DGS to reconstruct the scene and render novel trajectories. Then, DriveRestorer (Ni et al., 2024) mitigates artifacts in the rendered videos, and the results are used to fine-tune the reconstruction model. This

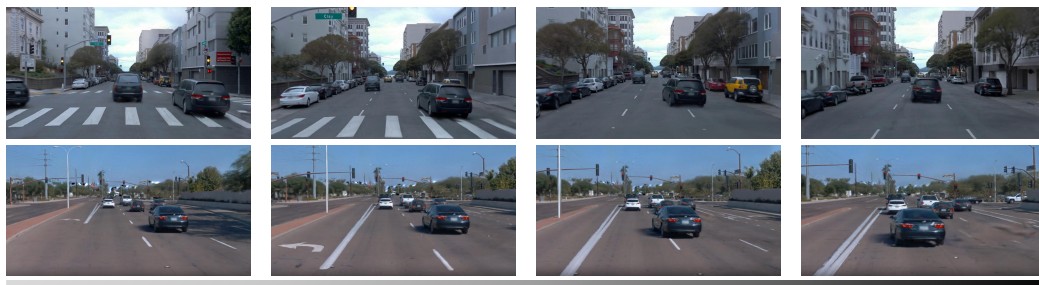

Figure 4: Examples of Dynamic Adversary Agent (DAA) controlling surrounding vehicles to simulate cut-in scenarios.

enables the final model to produce high-quality rendering from diverse viewpoints. The overall process is illustrated in Fig. 2. Meanwhile, we model the static background and moving vehicles separately, which enables the modification of trajectories and appearances for all vehicles. The background model is represented by $G_{\text{Background},w}$ in the world coordinate system. For each moving object $v$, represented by Gaussians $G_{\text{Rigid},l}^v$ in the local coordinate system $l$, these Gaussians must be transformed into the background coordinate system during the rendering process:

$$G_{\text{Rigid},w}^v(t) = M_t^v \cdot G_{\text{Rigid},l}^v + S_t^v, \tag{3}$$

where $M_t^v$ and $S_t^v$ represent the rotation matrix and translation vector that describe the pose of object $v$ at time $t$. This formulation ensures that moving vehicles can be accurately placed in relation to the static background as well as other dynamic agents during view rendering.

**Physical Modeling.** To maintain the authenticity of vehicle trajectories, a kinematic model is used to govern the vehicle's motion. Specifically, the vehicle's pose at time $t$ in the world coordinate system is $W_t = [R_t \mid P_t] \in SE(3)$, where $R_t$ is the rotation matrix and $P_t$ is the position. At each time step, the pose is updated using linear velocity $v_t$ and steering angle $\delta_t$ in a kinematic bicycle model:

$$P_{t+1} = P_t + v_t \cdot \Delta t \cdot \hat{d}_t, \tag{4}$$

where $\hat{d}_t$ is the forward direction vector derived from the rotation matrix $R_t$. The vehicle orientation is updated by applying a rotation about the vertical axis:

$$R_{t+1} = \text{Rot}_z(\Delta\theta_t) \cdot R_t, \tag{5}$$

where the incremental rotation angle $\Delta\theta_t$ is calculated based on the bicycle model as:

$$\Delta\theta_t = \frac{v_t}{L} \tan(\delta_t)\Delta t. \tag{6}$$

The $L$ denotes the wheelbase length of the vehicle. The rotation matrix $\text{Rot}_z(\Delta\theta_t)$ is explicitly expressed as:

$$\text{Rot}_z(\Delta\theta_t) = \begin{bmatrix} \cos(\Delta\theta_t) & \sin(\Delta\theta_t) & 0 \\ -\sin(\Delta\theta_t) & \cos(\Delta\theta_t) & 0 \\ 0 & 0 & 1 \end{bmatrix}. \tag{7}$$

We define different kinematic parameters for each vehicle category and perform checks when updating the trajectories to ensure the trajectory updates remain physically plausible and within the vehicle's operational constraints, including the maximum steering angle and velocity.

### 3.4 DYNAMIC ADVERSARY AGENT

Existing end-to-end autonomous driving models are typically trained with limited exposure to challenging corner cases. To tackle this limitation, we propose the Dynamic Adversary Agent (DAA), which generates highly realistic and diverse corner case interactions. The overview of DAA is provided in Fig. 3, and as illustrated in Fig. 4, it can effectively generate complex cut-in scenarios.

DAA first identifies suitable interactive target vehicles from the Bird's-Eye View (BEV) perspective, based on the distances to the ego vehicle and the specified interactive behavior $\mathcal{B}$. Different interactive behaviors correspond to different distance settings, which refer to traditional traffic flow

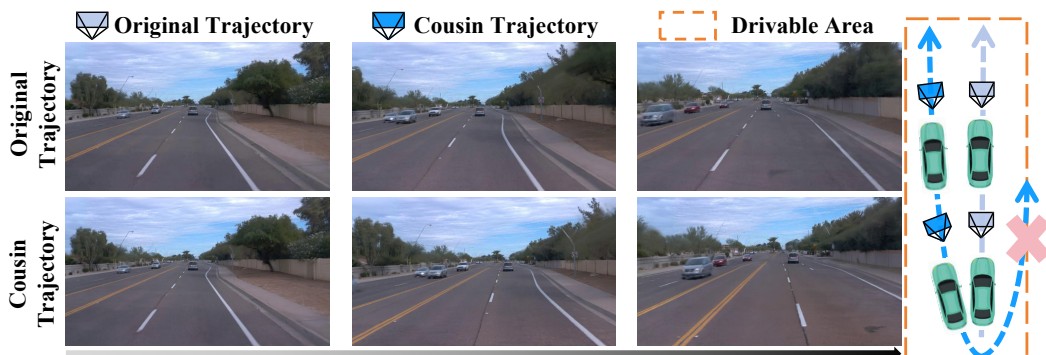

Figure 5: Cousin Trajectory Generator (CTG) generates cousin trajectories and performs trajectory checks to eliminate unreasonable trajectories (e.g., the pink cross marks), and finally renders the corresponding sensor data in the ReconSimulator.

methods (Kesting & Treiber, 2013; Treiber & Kesting, 2013). Then, a new trajectory is generated for the selected target vehicle by modifying its original trajectory $T_{\text{target}}$ based on both the ego vehicle's trajectory $T_{\text{ego}}$ and $\mathcal{B}$. The new trajectory $T'_{\text{target}}$ can be represented as:

$$T'_{\text{target}} = f(T_{\text{ego}}, T_{\text{target}}, \mathcal{B}), \tag{8}$$

where $f(\cdot)$ represents the trajectory generation function, which can be implemented using methods such as text-to-trajectory (Zhao et al., 2024b) and customized based on specific requirements. Then, the generated trajectory is checked for feasibility. First, the vehicle trajectory $T'_{\text{target}}$ should remain within the drivable region, and avoid collisions with other vehicles (collisions with the ego-vehicle are allowed):

$$\|T'_{\text{target}} - o_j\| \geq d_{\min}, \quad \forall j \in \{1, \ldots, M\}, \tag{9}$$

where $d_{\min}$ is the minimum distance between different agents $\{o_j\}_{j=1}^M$. Then, the trajectory is further checked to meet the constraints of the kinematic model and converted into the BEV perspective to verify whether it satisfies the interactive behavior $\mathcal{B}$.

DAA can be used in both imitation learning and reinforcement learning stages. In the first stage, DAA generates corner case scenarios by controlling surrounding vehicles and generating the ego vehicle's trajectory to avoid collisions, using a similar method mentioned above. The ReconSimulator then renders offline autonomous driving data based on the trajectories for imitation learning. In the second stage, the reinforcement learning policy controls the actions of the ego vehicle, while DAA controls the trajectory of the target vehicle in ReconSimulator to create corner cases. To enhance robustness, DAA has a certain probability of fine-tuning the trajectories (e.g., adjusting the target vehicle's speed) instead of directly reusing them from the first stage.

### 3.5 COUSIN TRAJECTORY GENERATOR

To address cold-start issues in reinforcement learning (Hua et al., 2021; Rashidinejad et al., 2021), behavior cloning is an effective pretraining strategy but requires large-scale real-world data, which are often biased toward simple scenarios (Codevilla et al., 2019; Zheng et al., 2024b; Sonmez et al., 2025). We propose the Cousin Trajectory Generator (CTG) to enhance action diversity, thereby creating the Cousin-nuScenes dataset with diverse actions.

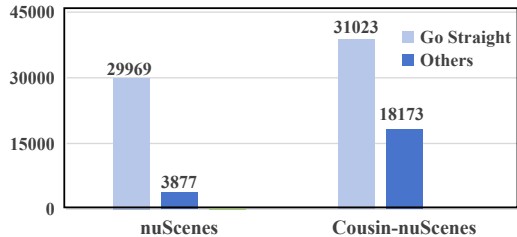

Figure 6: CTG creates Cousin-nuScenes with more diverse actions than nuScenes.

For trajectory extension, we generate new trajectories for the ego vehicle, such as lane changes and sharp turns, following a process similar to that described in Equation 8. Subsequently, a trajectory check is conducted to evaluate the validity of the generated trajectories, ensuring they remain physically plausible and adhere to the vehicle's operational limitations, including steering angle and

| Method | CR ↓ | DCR↓ | SCR↓ | DR↓ | PDR↓ | HDR↓ |
|---|---|---|---|---|---|---|
| VAD (Jiang et al., 2023) | 0.386 | 0.234 | 0.152 | 0.163 | 0.103 | 0.060 |
| GenAD (Zheng et al., 2024a) | 0.333 | 0.190 | 0.143 | 0.146 | 0.093 | 0.053 |
| VADv2 (Chen et al., 2024a) | 0.290 | 0.162 | 0.128 | 0.154 | 0.107 | 0.047 |
| RAD (Gao et al., 2025) | 0.238 | 0.143 | 0.095 | 0.084 | 0.057 | 0.027 |
| *ReconDreamer-RL* | **0.077** | **0.048** | **0.029** | **0.040** | **0.027** | **0.013** |

Table 1: Comparison of different methods on various metrics.

velocity, and verifying collision avoidance from the BEV perspective. Meanwhile, to better leverage expert trajectories from rare scenarios like U-turns, we interpolate the expert data for more detailed driving information. Given expert trajectories $\mathbf{X}_{\text{ego}} = \{X_{\text{ego}}^{t_1}, X_{\text{ego}}^{t_2}, \dots, X_{\text{ego}}^{t_n}\}$, where each $X_{\text{ego}}^{t_i}$ is the ego vehicle's position at time $t_i$, we perform linear interpolation between consecutive time steps $t_i$ and $t_{i+1}$. For an interpolated point $X_{\text{ego}}^t$ at time $t$, where $t_i \leq t \leq t_{i+1}$, the formula is:

$$X_{\text{ego}}^t = X_{\text{ego}}^{t_i} + \frac{t - t_i}{t_{i+1} - t_i}\left(X_{\text{ego}}^{t_{i+1}} - X_{\text{ego}}^{t_i}\right). \tag{10}$$

The time $t$ is expressed as:

$$t = t_i + k \cdot \Delta t, \quad k \in \{1, 2, \dots, m\}, \tag{11}$$

where $\Delta t = \frac{t_{i+1} - t_i}{m+1}$ and $m$ is the number of interpolation points between two time steps. For each interpolated trajectory point $X_{\text{ego}}^t$, we accordingly adjust the positions of surrounding vehicles to ensure the interpolated trajectories maintain a realistic spatial relationship and interactions.

After generating the interpolation and extension trajectories, we use the ReconSimulator to obtain the corresponding sensor data, which is then used for imitation training. We provide examples of trajectory extension in Fig. 5, illustrating two cases of ego-vehicle lane-changing maneuvers. Moreover, as shown in Fig. 6, Cousin-nuScenes created by CTG contains 4× more non-straight-line driving maneuvers than the nuScenes (Caesar et al., 2020) dataset.

## 4 EXPERIMENTS

In this section, we present our experimental setup, detailing the implementation specifics, the baseline used for evaluation, and the evaluation metrics. Subsequently, we present both quantitative and qualitative results, highlighting the enhanced performance achieved by *ReconDreamer-RL*. Finally, we perform ablation studies to assess the individual contributions of each component.

### 4.1 EXPERIMENTAL SETUP

**Implementation Details.** To demonstrate the effectiveness of the *ReconDreamer-RL* framework in enhancing end-to-end training, we employ it to train the autonomous driving policy in RAD (Gao et al., 2025). The overall training consists of two complementary stages: imitation learning and reinforcement learning. In the first stage, the perception capabilities are first strengthened, followed by planning pre-training through supervised behavior cloning. In the second stage, reinforcement learning is conducted within closed-loop 3DGS environments to further optimize the planning policy through interactive trial-and-error. All 3DGS environments are reconstructed from the nuScenes dataset (Caesar et al., 2020), and additional implementation details as well as examples of the edited scenes are provided in the supplementary material.

**Baseline.** In comparative analysis, we select representative end-to-end autonomous driving methods. For imitation learning, we choose VAD (Jiang et al., 2023), GenAD (Zheng et al., 2024a), and VADv2 (Chen et al., 2024a). For reinforcement learning, we use RAD (Gao et al., 2025), which is trained in closed-loop 3DGS environments.

**Evaluation Metrics.** Following the methodology described in RAD (Gao et al., 2025), we assess the performance of our policy using six distinct metrics. First, the Dynamic Collision Ratio (DCR) and Static Collision Ratio (SCR) are used to quantify the number of collisions with dynamic and

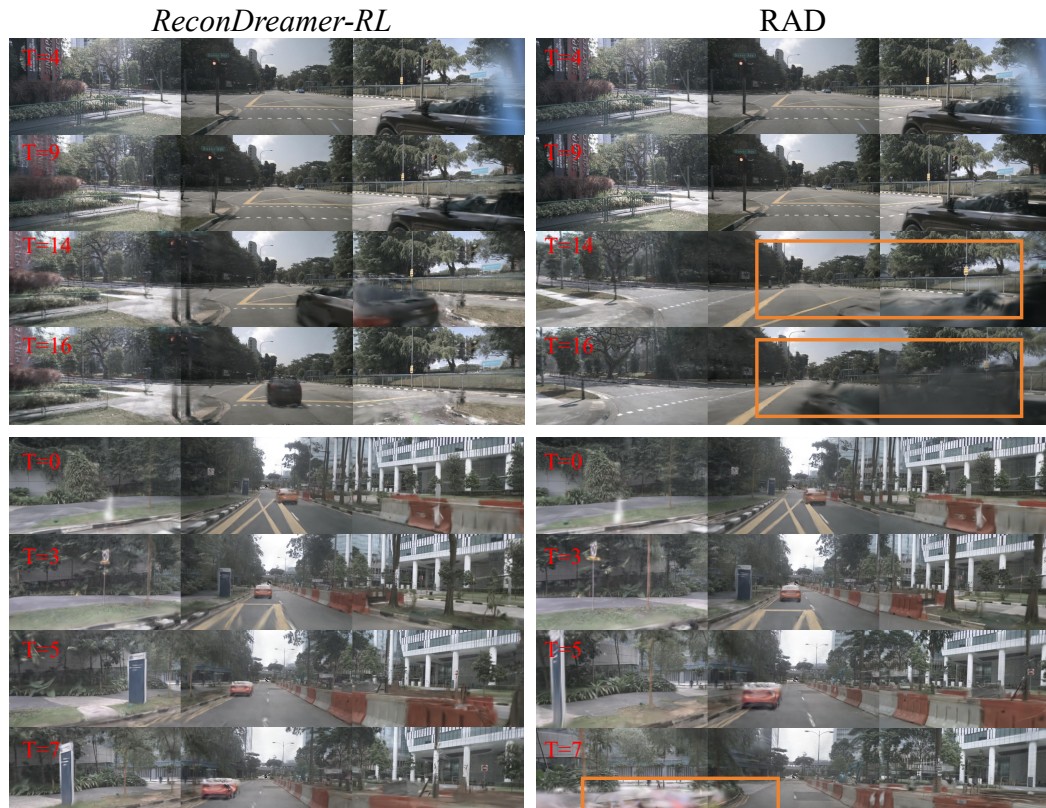

*ReconDreamer-RL*                                RAD

Figure 7: Comparison of different methods in challenging corner cases, with collisions highlighted by orange boxes.

| Simulator | EmerNeRF | RAD-3DGS | ReconSimulator |
|---|---|---|---|
| Speed (FPS) | 0.21 | 135 | 125 |

Table 2: Comparison of rendering speeds among reinforcement learning simulators, including EmerNeRF (Yang et al., 2023b), RAD (Gao et al., 2025), and ReconSimulator.

| Method | CR↓ | DCR↓ | SCR↓ |
|---|---|---|---|
| VAD (Jiang et al., 2023) | 0.449 | 0.293 | 0.156 |
| GenAD (Zheng et al., 2024a) | 0.379 | 0.234 | 0.145 |
| VADv2 (Chen et al., 2024a) | 0.436 | 0.276 | 0.160 |
| RAD (Gao et al., 2025) | 0.317 | 0.210 | 0.107 |
| *ReconDreamer-RL* | **0.089** | **0.053** | **0.036** |

Table 3: Collision metrics in cut-in scenarios.

static obstacles, respectively. The sum of these two values is referred to as the Collision Ratio (CR), which provides an overall measure of the policys ability to avoid obstacles in the environment. The Positional Deviation Ratio (PDR) and Heading Deviation Ratio (HDR) measure the ego vehicles deviation from the expert trajectory for position and heading, summed up and referred to as the Deviation Ratio (DR). Additionally, to evaluate comfort, we use Longitudinal Jerk (Long. Jerk) and Lateral Jerk (Lat. Jerk), which measure the rate of change in acceleration along the longitudinal and lateral axes, respectively. These metrics provide an assessment of the driving smoothness by capturing how abruptly the acceleration changes in each direction. To assess the realism of the reinforcement learning environment, we use three metrics from DriveDreamer4D (Zhao et al., 2024a): Novel Trajectory Agent IoU (NTA-IoU) for foreground quality, Novel Trajectory Lane IoU (NTL-IoU) for evaluating lane markings, and Frchet Inception Distance (FID) (Heusel et al., 2017) to quantify overall rendering fidelity.

## 4.2 MAIN RESULTS

**Quantitative Results.** *ReconDreamer-RL* enhances the performance of RAD (Gao et al., 2025) across all evaluation metrics, as shown in Tab. 1. Traditional imitation learning-based methods, such

| Method | Lane Shift @ 3m | | | Lane Shift @ 6m | | | Lane Change | | |
|---|---|---|---|---|---|---|---|---|---|
| | NTA-IoU ↑ | NTL-IoU ↑ | FID ↓ | NTA-IoU ↑ | NTL-IoU ↑ | FID ↓ | NTA-IoU ↑ | NTL-IoU ↑ | FID ↓ |
| w/o Video Diffusion Prior | 0.224 | 49.67 | 160.23 | 0.148 | 44.89 | 256.42 | 0.215 | 46.23 | 178.34 |
| w/ Video Diffusion Prior | **0.342** | **50.12** | **130.48** | **0.325** | **49.72** | **125.43** | **0.337** | **50.01** | **122.67** |

Table 4: Impact of diffusion prior within ReconSimulator on novel view rendering quality across lane shift and lane change.

| Method | Long. Jerk↓ | Lat. Jerk↓ |
|---|---|---|
| VAD (Jiang et al., 2023) | 5.645 | 0.597 |
| GenAD (Zheng et al., 2024a) | 12.490 | 0.432 |
| VADv2 (Chen et al., 2024a) | 8.124 | 0.435 |
| RAD (Gao et al., 2025) | **4.237** | 0.191 |
| *ReconDreamer-RL* | 4.264 | **0.189** |

Table 5: Comfort analysis in unedited 3DGS environments.

| Method | Long. Jerk↓ | Lat. Jerk↓ |
|---|---|---|
| VAD (Jiang et al., 2023) | 6.714 | 0.784 |
| GenAD (Zheng et al., 2024a) | 14.216 | 0.548 |
| VADv2 (Chen et al., 2024a) | 9.245 | 0.499 |
| RAD (Gao et al., 2025) | 4.912 | 0.316 |
| *ReconDreamer-RL* | **3.832** | **0.276** |

Table 6: Comfort analysis in unedited in cut-in scenarios.

as VAD (Jiang et al., 2023) and VADv2 (Chen et al., 2024a), struggle with closed-loop evaluation tasks due to the gap between training and closed-loop inference. RAD (Gao et al., 2025) enhancing and training within a 3DGS-based simulator. However, due to limited corner case coverage and poor novel view rendering ability of 3DGS, RAD exhibits a higher collision rate in our challenging evaluation benchmark. However, *ReconDreamer-RL* leverages diffusion prior and data augmentation to enhance end-to-end autonomous driving, achieving a 3× reduction in Collision Ratio compared to RAD (Gao et al., 2025).

**Corner Case Results.** As shown in Tab. 3, we compare the collision metrics of different methods in corner cases, focusing on cut-in situations. Imitation learning-based methods exhibit a high Dynamic Collision Ratio in cut-in corner cases, as such scenarios are completely absent from the training data. RAD (Gao et al., 2025) improves the model's ability to handle corner cases through reinforcement learning, but still lacks the necessary data to train on such situations. In contrast, *ReconDreamer-RL* improves the policy's performance through a two-stage process, achieving 404.5% improvement in Collision Ratio over imitation methods.

**Rendering Speed.** Rendering speed is crucial for reinforcement learning environments. As shown in Tab. 2, ReconSimulator achieves an efficient rendering speed of 125 FPS, which is significantly faster than EmerNeRF's 0.21 FPS and comparable to RAD-3DGS's 135 FPS. This high rendering speed enables ReconSimulator to satisfy the demanding requirements of reinforcement learning.

**Qualitative Results.** As shown in Fig. 7, we compare *ReconDreamer-RL* and RAD Gao et al. (2025) under two corner cases. The first scenario involves a vehicle on the right of the ego car quickly cutting in. In this case, *ReconDreamer-RL* successfully avoids the collision and maintains a safe distance, while RAD (Gao et al., 2025) fails to react in time, leading to a potential collision due to inadequate planning. The second scenario features a vehicle on the right performing a cut-in, followed by a sudden brake. Although RAD (Gao et al., 2025) attempts to avoid the collision by shifting lanes, it fails to control its speed and steering angle properly, resulting in a less effective response. In contrast, *ReconDreamer-RL* not only safely changes lanes but also decelerates smoothly, ensuring that the ego vehicle avoids the collision while maintaining a stable and controlled trajectory.

**Comfort Results.** As shown in Table 5, we compare the comfort analysis in unedited 3DGS environments. We observe that *ReconDreamer-RL* performs similarly to RAD (Gao et al., 2025) in terms of Long. Jerk and Lat. Jerk metrics, and significantly outperforms algorithms that rely solely on imitation learning. This is because *ReconDreamer-RL* employs a strategy that alternates between imitation learning and reinforcement learning during training, as detailed in Appendix D. This approach allows *ReconDreamer-RL* to not only improve safety but also better align with human driving behavior. Additionally, we compare comfort metrics in cut-in scenarios, evaluating only the scenes where no collisions occur across all algorithms. We find that *ReconDreamer-RL* consistently delivers the highest performance in these scenarios, as it learns more effective collision avoidance strategies from a large set of corner cases, whereas other methods lack such data, leading to instability when encountering these corner cases.

| ReconSimulator | DAA | CTG | CR↓ | DCR↓ | SCR↓ | DR↓ | PDR↓ | HDR↓ |
|---|---|---|---|---|---|---|---|---|
| | | | 0.238 | 0.143 | 0.095 | 0.084 | 0.057 | 0.027 |
| | ✓ | | 0.167 | 0.102 | 0.065 | 0.076 | 0.052 | 0.024 |
| | | ✓ | 0.191 | 0.121 | 0.070 | 0.068 | 0.046 | 0.022 |
| | ✓ | ✓ | 0.142 | 0.082 | 0.060 | 0.063 | 0.043 | 0.020 |
| ✓ | | | 0.172 | 0.103 | 0.069 | 0.073 | 0.053 | 0.020 |
| ✓ | ✓ | | 0.117 | 0.069 | 0.048 | 0.067 | 0.050 | 0.017 |
| ✓ | | ✓ | 0.143 | 0.086 | 0.057 | 0.053 | 0.040 | 0.013 |
| ✓ | ✓ | ✓ | **0.077** | **0.048** | **0.029** | **0.040** | **0.027** | **0.013** |

Table 7: Ablation study of *ReconDreamer-RL*, evaluating the effectiveness of each module. When ReconSimulator is not used, we directly employ the RAD-3DGS (Gao et al., 2025) method to reconstruct the scenes.

## 4.3 ABLATION STUDIES

**ReconSimulator.** As shown in Tab. 4, we conduct ablation experiments on ReconSimulator to demonstrate the impact of the diffusion prior on novel view rendering quality. Specifically, without ReconSimulator, the NTA-IoU and NTL-IoU drop by 54.46% and 9.71%, respectively, while the FID metric increases by 104.43% for a 6m lane shift, indicating significant degradation in the appearance of surrounding vehicles and lane markings. This suggests that the absence of ReconSimulator affects the visual consistency of the rendered scenes. Meanwhile, in rows 2, 3, and 4 of Tab. 7, we use RAD-3DGS to generate DAA and CTG data, instead of ReconSimulator, to

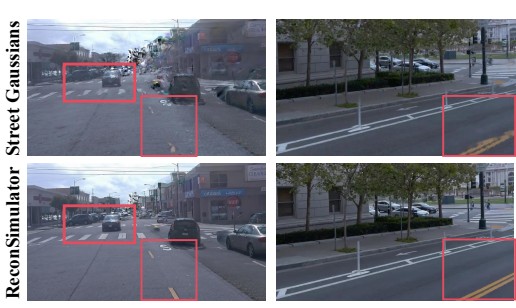

Figure 8: Comparison of ReconSimulator and Street Gaussians in rendering novel views.

train the RAD policy, while keeping the rendering process unchanged. Despite the lower rendering quality of RAD-3DGS, performance still improves due to the absence of corner cases present in the original data. As shown in rows 6, 7, and 8 of Tab. 7, using ReconSimulator to generate CTG and DAA data results in more significant performance gains. We also compare ReconSimulator with traditional 3DGS, which separates foreground and background. As shown in the red box of Fig. 8, ReconSimulator yields sharper lane markings and a cleaner background during policy exploration.

**Impact of the DAA.** The DAA helps reduce collision occurrences by generating corner case scenarios for training. As shown in Tab. 7, incorporating DAA decreases the Collision Ratio from 0.172 to 0.117, demonstrating its effectiveness in improving model safety and reliability.

**Impact of the CTG.** In Tab. 7, the benefit of CTG is validated by reductions in CR (from 0.172 to 0.143) and DR (from 0.073 to 0.052), indicating improved trajectory adherence. By addressing the data bias, CTG enhances the action distribution learned during the imitation learning phase.

## 5 CONCLUSION

This paper introduces *ReconDreamer-RL*, a novel and comprehensive framework designed to enhance reinforcement learning for end-to-end autonomous driving through the integration of video diffusion priors. It focuses on constructing a more realistic simulator, enriching corner cases, and addressing the issue of training data distribution. Specifically, we introduce ReconSimulator, which combines the video diffusion prior for appearance modeling and a kinematic model for physical dynamics, providing realistic environments for reinforcement learning. Additionally, we develop the DAA, which generates complex traffic scenarios, such as sudden lane changes. To address the issue of training data distribution being mostly on simple straight-line movements, CTG is proposed to enhance trajectory diversity through extension and interpolation. Experimental evaluations indicate that *ReconDreamer-RL* significantly improves the performance of end-to-end autonomous driving models, reducing the Collision Ratio by 5× compared to imitation-based learning methods.

## 6 REPRODUCIBILITY STATEMENT

The details of our method can be found in Section 3 and the appendix, which outline specific parameter configurations and implementation particulars. We also provide an in-depth overview of the datasets used, covering their origins, characteristics, and any preprocessing steps taken to ensure data integrity. Furthermore, we explore the rationale behind selecting these specific datasets and explain how they align with the objectives of our research. To promote reproducibility, all associated code and datasets will be made publicly accessible following publication. This will enable other researchers to easily replicate our experiments, verify our results, and build upon our work. We believe that maintaining transparency in research practices is essential for advancing knowledge in our field and fostering collaborative efforts among researchers.

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

## A  DATASETS

We reconstruct the entire nuScenes (Caesar et al., 2020) and Waymo (Sun et al., 2020) datasets into 3D Gaussian Splatting (3DGS) (Kerbl et al., 2023) environments. These reconstructed environments will be made publicly available to support future research.

**nuScenes.** The nuScenes (Caesar et al., 2020) dataset is a large-scale benchmark designed for the evaluation of autonomous driving systems, comprising a diverse collection of urban driving scenarios. It provides approximately 1.4 million annotated 3D bounding boxes across 23 distinct object categories, offering rich and detailed annotations for various objects encountered in complex driving environments. The dataset captures scene imagery using six high-resolution cameras, which together provide a full 360-degree horizontal field of view, ensuring comprehensive coverage of the vehicle's surroundings. Keyframes are annotated at a frequency of 2 Hz, allowing for detailed temporal tracking of dynamic objects.

**Waymo.** To further evaluate the generalization capability of *ReconDreamer-RL*, we extend our experiments to the Waymo (Sun et al., 2020) dataset, a widely recognized and comprehensive benchmark in the autonomous driving field. The Waymo dataset features a wide variety of complex urban driving scenarios, encompassing different traffic conditions, road types, and environmental factors such as lighting, weather, and pedestrian activity. This diversity allows for a robust evaluation of model performance across real-world situations, ensuring that our approach can generalize effectively in dynamic and unpredictable settings.

## B  METRIC

To evaluate the performance of the autonomous driving policy, we use six key metrics following RAD (Gao et al., 2025).

The Dynamic Collision Ratio (DCR) represents the frequency of collisions with dynamic obstacles and is calculated as:

$$\text{DCR} = \frac{N_{\text{dc}}}{N_{\text{total}}}, \tag{12}$$

where $N_{\text{dc}}$ denotes the number of clips where collisions with dynamic obstacles occur, and $N_{\text{total}}$ represents the total number of clips.

The Static Collision Ratio (SCR) quantifies the occurrence of collisions with static obstacles and is defined as:

$$\text{SCR} = \frac{N_{\text{sc}}}{N_{\text{total}}}, \tag{13}$$

where $N_{\text{sc}}$ is the number of clips with static obstacle collisions.

The Collision Ratio (CR) represents the overall collision frequency and is given by:

$$\text{CR} = \text{DCR} + \text{SCR}. \tag{14}$$

The Positional Deviation Ratio (PDR) measures how closely the ego vehicle follows the expert trajectory in terms of position. It is defined as:

$$\text{PDR} = \frac{N_{\text{pd}}}{N_{\text{total}}}, \tag{15}$$

where $N_{\text{pd}}$ denotes the number of trajectory clips in which the positional deviation exceeds a predefined threshold.

The Heading Deviation Ratio (HDR) evaluates orientation accuracy by calculating the proportion of clips in which the heading deviation exceeds a predefined threshold.

$$\text{HDR} = \frac{N_{\text{hd}}}{N_{\text{total}}}, \tag{16}$$

where $N_{\text{hd}}$ is the number of clips where the heading deviation exceeds the threshold.

The Deviation Ratio (DR) provides a combined metric of positional and orientation deviations from the expert trajectory, formulated as:

$$\text{DR} = \text{PDR} + \text{HDR}. \tag{17}$$

Finally, Longitudinal Jerk (denoted as Long. Jerk) and Lateral Jerk (denoted as Lat. Jerk) are used to assess the smoothness of a vehicle's motion by quantifying the changes in acceleration. The longitudinal jerk, which measures the rate of change of the vehicle's longitudinal acceleration, is given by:

$$\text{Long. Jerk} = \frac{d^2 v_{long}}{dt^2} \tag{18}$$

where $v_{\text{long}}$ represents the longitudinal velocity of the vehicle. Similarly, the lateral jerk, which measures the rate of change of the vehicle's lateral acceleration, is defined as:

$$\text{Lat. Jerk} = \frac{d^2 v_{lat}}{dt^2} \tag{19}$$

where $v_{\text{lat}}$ denotes the lateral velocity. These two metrics help in capturing abrupt changes in both acceleration and steering, offering a comprehensive understanding of the comfort level for passengers and the stability of the vehicle during driving.

Meanwhile, since the edited scenes lack ground-truth expert trajectories, we only compute the collision-related metrics for the scenarios. And, we adopt Novel Trajectory Agent IoU (NTA-IoU) and Novel Trajectory Lane IoU (NTL-IoU) from DriveDreamer4D (Zhao et al., 2024a) to evaluate the realism of the environment in reinforcement learning, as they are specifically designed to measure the spatiotemporal consistency of dynamic agents (foreground) and static lane structures (background) in rendered videos.

The NTA-IoU method utilizes the YOLOv11 detector (Khanam & Hussain, 2024) to extract 2D bounding boxes of vehicles from rendered images that are generated along novel trajectories. This step ensures that vehicle locations are accurately represented in the new perspective. Simultaneously, the original 3D bounding boxes are transformed into the coordinate frame of the novel trajectory. This transformation allows the 3D boxes to be correctly projected into 2D bounding boxes under the novel viewpoint. For each projected 2D box, NTA-IoU identifies the closest detected box from the YOLOv11 model and calculates the Intersection over Union (IoU) between the projected and detected bounding boxes. The IoU score serves as a measure of the accuracy and overlap between the projected 2D boxes and the corresponding detected boxes, providing a quantitative assessment of the model's ability to match the projected vehicle locations with those detected in the new viewpoint.

In a similar fashion, the NTL-IoU is computed by first performing geometric transformations to project the lane structures from the original trajectories into the viewpoint of the novel trajectories. This step ensures that the lane structures are properly aligned in the new perspective. Simultaneously, lane lines in the newly rendered images are extracted using the TwinLiteNet model (Che et al., 2023), which is specifically designed for lane detection. Once the lane lines are extracted, the mean Intersection over Union (IoU) is calculated by comparing the projected lane lines with those detected by TwinLiteNet. The IoU score quantifies the accuracy of the lane line projections relative to the detected lanes, providing a measure of the alignment between the two sets of lane lines.

## C    DETAILS OF APPEARANCE MODELING OF RECONSIMULATOR

The core of appearance modeling in ReconSimulator is to integrate a diffusion prior to enhance scene representation and enable high-quality rendering of sensor data along novel trajectories. This process occurs in two processes. The first process involves training DriveRestorer, a diffusion model designed to mitigate artifacts. In the second process, the trained model is applied to enhance the appearance of 4D scene reconstructions.

**Training Details for DriveRestorer.** For training DriveRestorer, we first leverage under-trained reconstruction models to render videos along the original trajectory. Due to model underfitting, ghosting artifacts naturally appear in the rendered videos. These degraded frames are paired with their corresponding ground truth videos, forming a restoration dataset. Specifically, it is fine-tuned based on video diffusion models (Wang et al., 2023; Zhao et al., 2024b) and uses a diffusion loss:

$$\mathcal{L}_{\mathcal{R}} = \mathbb{E}_{\boldsymbol{z}, \epsilon_t \sim \mathcal{N}(0,1), t} \left[ \left\| \epsilon_t - \epsilon_\theta \left( \boldsymbol{z}_t, t, s \right) \right\|_2^2 \right], \tag{20}$$

where $\epsilon_t$ is the random noise at time step $t$, $\epsilon_\theta$ is the denoising network used to predict the noise, $\boldsymbol{z}_t$ is the noisy latent variable at step $t$, and $s$ represents the control conditions (including degraded video, 3D boxes, and HDMaps).

| Model | NTA-IoU ↑ | NTL-IoU ↑ | FID ↓ | Extra Diffusion Training Time (h) | Single Reconstruction Time (h) |
|---|---|---|---|---|---|
| RAD-3DGS (Gao et al., 2025) | 0.219 | 48.72 | 149.50 | – | 1.5 |
| ReconSimulator w/o finetune | 0.287 | 49.75 | 144.31 | – | 1.8 |
| ReconSimulator w/ finetune | 0.317 | **49.84** | 127.10 | 3.0 | 1.8 |
| ReconSimulator re-trained | **0.325** | 49.72 | **125.43** | 12.0 | 1.8 |

Table 8: Computational overhead and generalization performance of ReconSimulator.

**The process of reconstructing scene with DriveRestorer.** The scene reconstruction process with DriveRestorer is divided into two stages. In the first stage, we train the scene reconstruction model $G$ using the original data $V_{ori}$ for $k_1$ steps, until the rendered quality reaches an acceptable level from the perspective of $\mathcal{T}_{ori}$. During this stage, the reconstruction loss is:

$$Loss = \lambda_1 L_{RGB}^{ori} + \lambda_2 L_{Depth}^{ori} + \lambda_3 L_{SSIM}^{ori}. \tag{21}$$

In the second stage, the trained model $G$ is first used to render novel trajectory videos $\hat{V}_{novel}$:

$$\hat{V}_{novel} = G(\mathcal{T}_{novel}), \tag{22}$$

where $\mathcal{T}_{novel}$ is the novel trajectory. These rendered videos $\hat{V}_{novel}$ often exhibit ghost artifacts due to data sparsity. Then, DriveRestorer freezes its network parameters and is introduced to restore these artifacts. The inference process can be expressed as:

$$V_{novel} = \mathcal{R}(\hat{V}_{novel}, c), \tag{23}$$

where $\mathcal{R}$ represents the DriveRestorer model, and $c$ denotes the structural conditions corresponding to the degraded video $\hat{V}_{novel}$. A mixed dataset $V$ is then constructed by combining the original trajectory video dataset $V_{ori}$ and the restored novel trajectory video dataset $V_{novel}$. This mixed dataset is used to further train the scene reconstruction model $G$ for $k_2$ steps:

$$V = 0.5 \cdot V_{ori} \cup 0.5 \cdot V_{novel}. \tag{24}$$

The further training on the dataset $V$ enables $G$ to render high-quality images across a broader range of trajectories. Meanwhile, to meet the requirements of reinforcement learning for free exploration of scenes, we gradually update $V_{novel}$. Specifically, during every $k_3$-th step of training $G$, a new $\mathcal{T}'_{novel}$ trajectory is selected. The reconstruction model $G$ then renders the new trajectory videos $\hat{V}'_{novel}$, which are processed by DriveRestorer to obtain the restored novel trajectory videos $V'_{novel}$. These videos are then added to the $V_{novel}$ dataset. To ensure that the newly generated data provides additional priors for the reconstruction model, the updated dataset $V_{novel}$ is modified as follows:

$$V_{novel} = (1 - w) \cdot V_{novel} \cup w \cdot V'_{novel}, \tag{25}$$

where $w$ is the sampling probability. In this stage, the loss function is:

$$Loss = \lambda_1 L_{RGB}^{ori} + \lambda_2 L_{Depth}^{ori} + \lambda_3 L_{SSIM}^{ori} + \lambda_4 L_{RGB}^{novel} + \lambda_5 L_{SSIM}^{novel}, \tag{26}$$

where $L_{RGB}^{ori}$ is the RGB reconstruction loss for the original dataset, $L_{Depth}^{ori}$ is the depth loss, $L_{SSIM}^{ori}$ is the structural similarity loss for the original dataset, $L_{RGB}^{novel}$ is the RGB reconstruction loss for the novel trajectory dataset, and $L_{SSIM}^{novel}$ is the structural similarity loss for the novel trajectory dataset. Notably, the synthetic dataset does not use depth as supervision.

**The Cost of integrating diffusion prior and the generalization ability of DriveRestorer .** To analyze the cost of integrating the diffusion prior and the generalization ability of DriveRestorer, we compare different strategies for the ReconSimulator. First, we train a DriveRestorer model on the Waymo dataset (Sun et al., 2020) and apply it directly to the ReconSimulator for reconstructing the nuScenes dataset (Caesar et al., 2020), assessing both computational overhead and generalization performance. As shown in Tab. 8, when the ReconSimulator utilizes the DriveRestorer model trained on the Waymo dataset for reconstruction, it still significantly improves reconstruction on nuScenes, demonstrating strong cross-dataset generalization. This generalization substantially reduces the cost of integrating the diffusion prior. Moreover, fine-tuning DriveRestorer on a target dataset is relatively inexpensive, requiring only 3 hours, whereas training from scratch takes 12 hours, which remains reasonable. Additionally, this overhead is incurred only once for reconstructing all the scenes in the target dataset. During the scene reconstruction process, the integration of the diffusion prior introduces minimal overhead, adding only about 0.3 hours per scene, as it only requires inference during reconstruction.

| Config | Imitation Learning | Reinforcement Learning |
|---|---|---|
| Learning rate | $1 \times 10^{-4}$ | $5 \times 10^{-6}$ |
| Learning rate schedule | Cosine decay | Cosine decay |
| Optimizer | AdamW (Loshchilov & Hutter, 2017) | AdamW (Loshchilov & Hutter, 2017) |
| Optimizer hyper-parameters | $\beta_1 = 0.9, \beta_2 = 0.999, \epsilon = 10^{-8}$ | $\beta_1 = 0.9, \beta_2 = 0.999, \epsilon = 10^{-8}$ |
| Weight decay | $1 \times 10^{-4}$ | $1 \times 10^{-4}$ |
| GAE parameters | - | $\gamma = 0.9, \lambda = 0.95$ |
| Clipping thresholds | - | $\epsilon_x = 0.1, \epsilon_y = 0.2$ |
| Planning head dimension | 256 | 256 |

Table 9: Hyperparameters for imitation learning and reinforcement learning.

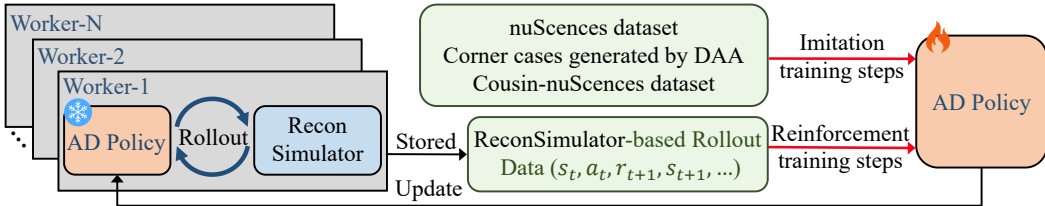

Figure 9: We initiate $N$ ReonSimulators, each corresponding to a random scenario. The policy then interacts with each ReconSimulator to gather reinforcement learning training data, which is subsequently stored. Meanwhile, we alternate between reinforcement learning and imitation learning to ensure stable learning.

## D  TRAINING DETAILS OF THE *ReconDreamer-RL*

To demonstrate the capability of the *ReconDreamer-RL* framework in enhancing training, we apply it to train RAD (Gao et al., 2025). Next, we introduce the network architecture of RAD (Gao et al., 2025) and its training stages. And, we summarize the detailed hyperparameter settings for both the imitation learning and reinforcement learning stages in Tab. 9.

**Network Architecture of RAD.** RAD (Gao et al., 2025) takes multi-view images as input, transforms sensor data into scene token embeddings, and outputs the probabilistic distribution of actions. The framework consists of a BEV encoder, map head, agent head, image encoder, and planning head. Specifically, the BEV encoder transforms multi-view features from the perspective view into birds-eye-view (BEV) representations (Li et al., 2024c; Yang et al., 2023a). The map head extracts vectorized road elements. The agent head predicts motion states and trajectories of surrounding dynamic agents. Finally, the planning head outputs the probabilistic distribution of driving actions. Meanwhile, RAD (Gao et al., 2025) employs a two-stage training approach, including the imitation learning stage and the reinforcement learning stage.

**Imitation Learning Stage.** In this stage, perception training is conducted, where the map head and agent head explicitly predict map elements and agent motion information, supervised by ground-truth labels. Subsequently, imitation learning is employed to initialize the action distribution based on large-scale driving demonstrations from expert drivers (Caesar et al., 2020; Sun et al., 2020; Xiao et al., 2021). The data sources for imitation learning in this stage consist of three components: first, demonstration data from the nuScenes dataset (Caesar et al., 2020), second, various corner cases created through the Dynamic Adversary Agent (DAA), and third, the Cousin-Nuscence dataset created by the Cousin Trajectory Generator (CTG). During this stage, only the parameters of the image encoder and planning head are updated, while the parameters of the BEV encoder, map head, and agent head remain frozen.

**Reinforcement learning stage.** In the reinforced post-training stage, we fine-tune the action distribution using both reinforcement learning and imitation learning in parallel. The goal of reinforcement learning is to guide the policy to be sensitive to critical events, such as collisions, and to adapt to various out-of-distribution situations. Imitation learning, on the other hand, aims to ensure that the policy's behavior closely resembles human behavior, thereby maintaining stability.

To collect data, we independently reconstruct each video clip in the dataset (Caesar et al., 2020; Sun et al., 2020) using ReconSimulator. As shown in Fig. 9, we utilize $N$ parallel workers. Each worker

| RL:IL | CR | DCR | SCR | DR | PDR | HDR |
|-------|-----|------|------|------|------|------|
| 0:1 | 0.238 | 0.143 | 0.095 | 0.084 | 0.057 | 0.027 |
| 1:0 | 0.111 | 0.069 | 0.042 | 0.057 | 0.037 | 0.020 |
| 2:1 | 0.091 | 0.056 | 0.035 | 0.055 | 0.034 | 0.021 |
| 4:1 | **0.077** | **0.048** | **0.029** | **0.040** | 0.027 | **0.013** |
| 8:1 | 0.090 | 0.058 | 0.032 | 0.042 | **0.025** | 0.017 |

Table 10: Ablation study on the impact of RL-to-IL step mixing ratios on final performance.

| Real Data | Synthetic Data | CR↓ | DR↓ |
|-----------|----------------|------|------|
| 100% | 0% | 0.150 | 0.076 |
| 80% | 20% | 0.114 | 0.052 |
| 60% | 40% | 0.061 | 0.032 |
| 40% | 60% | 0.053 | 0.021 |
| 20% | 80% | **0.038** | **0.018** |

| Real Data | Synthetic Data | CR↓ | DR↓ |
|-----------|----------------|------|------|
| 100% | 0% | 0.362 | 0.231 |
| 80% | 20% | 0.246 | 0.175 |
| 60% | 40% | 0.189 | 0.123 |
| 40% | 60% | 0.143 | 0.104 |
| 20% | 80% | **0.113** | **0.095** |

Table 11: The analyze of how the amount of synthetic data used influences model performance in nominal driving scenarios, while keeping the total amount of real data constant and only varying the quantity of synthetic data.

Table 12: The analysis of how the amount of synthetic data used influences model performance in cut-in scenarios, while keeping the total amount of real data constant and only varying the quantity of synthetic data.

randomly selects a ReconSimulator environment, where an autonomous driving policy governs the ego vehicle, enabling it to interact continuously with the environment to gather data. The resulting trajectory data, including the sequence $(s_t, a_t, r_{t+1}, s_{t+1}, \dots)$, is then stored. The trajectory data $(s_t, a_t, r_{t+1}, s_{t+1}, \dots)$ represents the agent's experience, where $s_t$ is the state at time $t$, $a_t$ is the action taken, $r_{t+1}$ is the reward received, and $s_{t+1}$ is the next state.

For policy optimization, we alternate between reinforcement learning and imitation learning training phases. During the reinforcement learning phase, we sample data from the collected dataset and apply the Proximal Policy Optimization (PPO) algorithm (Schulman et al., 2017) to update the policy. In the imitation learning phase, we leverage data from three sources: nuScence dataset, corner cases generated by ReconSimulator, and the Cousin-nuScenes dataset. After a predefined number of training steps, the updated policy is distributed to each worker, replacing the previous version. At this stage, only the image encoder and planning head parameters are updated, while the parameters of the BEV encoder, map head, and agent head remain fixed. We also analyze the effect of the ratio between reinforcement training and imitation training on the final results. Our findings show that a 4:1 ratio results in the lowest collision rate, as shown in Tab. 10.

| Scenario Count | CR↓ |
|----------------|------|
| 1000 | 0.172 |
| 2000 | 0.154 |
| 4000 | 0.131 |
| 8000 | 0.107 |
| 16000 | 0.096 |
| 32000 | **0.081** |

Table 13: Relationship between scenario count and final performance in synthetic adversarial scenarios.

We conduct a detailed analysis of the training strategy in both the imitation learning stage and the reinforcement learning stage. In the imitation learning stage, we study the effect of the ratio between real data (from the nuScenes dataset (Caesar et al., 2020)) and synthetic data (generated by DAA and CTG) on training performance. During this process, the amount of real data is kept fixed, and only the amount of synthetic data is varied. As shown in Tab. 11, we evaluate the imitation learning models in a closed-loop simulation environment using unedited closed-loop scenarios as the test set, and compare the performance of models trained with different proportions of augmented data. The results show that incorporating a mixed dataset con-

| Scenario Count | CR↓ | DR↓ |
|----------------|------|------|
| 1000 | 0.101 | 0.063 |
| 2000 | 0.092 | 0.051 |
| 4000 | 0.087 | 0.044 |
| 8000 | 0.081 | 0.032 |
| 16000 | 0.077 | 0.024 |
| 32000 | **0.071** | **0.023** |

Table 14: Relationship between scenario count and final performance the nominal real driving domain.

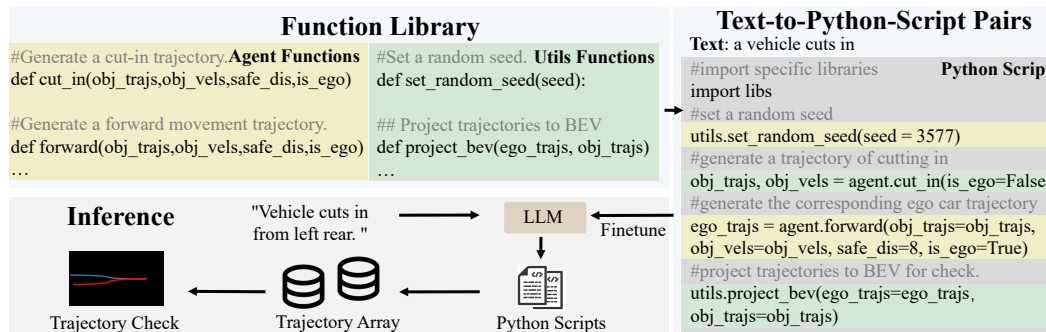

Figure 10: We first construct a function library, which includes various agent behaviors such as steering, acceleration, and braking. These functions are then utilized to create a dataset for fine-tuning a large language model (LLM) to automatically generate agent trajectories.

sisting of both real and synthetic data still brings noticeable performance gains on real-domain data with nominal driving scenarios. This improvement can be attributed to the ReconSimulator, which generates synthetic data whose distribution closely matches that of the real domain, thereby ensuring high-quality synthetic data. Meanwhile, we compare the collision metrics of different methods in corner-case scenarios, with a particular focus on cut-in situations as shown in Tab. 12. To directly assess the proximity of the model and the generated avoidance trajectory under corner cases, we perform DR calculations using the generated avoidance trajectory. We observe that methods leveraging synthetic data significantly outperform those that do not use synthetic data, and that the collision rate and related metrics consistently decrease as the amount of synthetic data increases.

Furthermore, we conducted an analysis to investigate how sample complexity impacts the final performance during training. To assess sample complexity, we use the scenario count, where each modification to the surrounding vehicle trajectories or the ego vehicles navigation (either altered by CTG or DAA) is considered a distinct scenario. Our experimental results indicate that an increase in the scenario count generally results in improved final performance. This trend is clearly illustrated in Tab. 13 and Tab. 14, which highlight the correlation between the number of scenarios and the performance outcomes.

## E  TRAJECTORY GENERATION AND CHECK IN DAA

**Trajectory Generation.** To enable the automated creation of more complex and diverse autonomous-driving scenarios, we first develop a comprehensive library of trajectory-generation functions. This library is then used to fine-tune a large language model (LLM), enabling direct translation of natural-language descriptions into various agent trajectories. As shown in Fig. 10, the library includes a variety of agent behaviours (e.g., steering, constant-speed driving, acceleration, and braking) as well as additional utilities, such as trajectory projection. Using these functions, we manually cu-

| Samples | Interesting Adversarial Scenarios | Percentage |
|---------|-----------------------------------|------------|
| 1000    | 435                               | 43.5%      |
| 2000    | 892                               | 44.6%      |
| 4000    | 1653                              | 41.3%      |
| 8000    | 3470                              | 43.4%      |
| 16000   | 6712                              | 42.0%      |

Table 15: Summary of Challenging Scenarios for Different Sample Sizes.

rate an extensive dataset of text-to-Python script pairs, which we then use to fine-tune the LLM. The fine-tuned LLM is then leveraged to generate corner cases and other driving scenarios, which are discussed in detail in Appendix F. As shown in Tab. 15, we present the proportion of interesting adversarial scenarios in the total sample. These scenarios are defined as those in which we use the frozen-parameter policy trained during the imitation learning phase to check whether the predictions in edited scenarios differ from those in unedited ones. The prediction window is within 0.5 seconds, and a scenario is considered an interesting adversarial scenario if the deviation exceeds 0.5 meters in the lateral direction or 1 meter in the longitudinal direction. Additionally, a scenario is also con-

sidered relevant if the lateral and longitudinal distances between the selected target vehicle and the ego vehicle are below the corresponding thresholds1 meter in the lateral direction and 2 meters in the longitudinal direction. As shown in Tab. 15, DAA exhibits a high ratio of interesting adversarial scenarios. This can also be attributed to the imposed constraints. For example, when identifying potentially interactive target vehicles, such as a vehicle cutting in from the left rear, this significantly reduces the number of feasible interactions based on direction. Furthermore, vehicle types must adhere to physical laws regarding speed and other characteristics, which further restricts the set of vehicles capable of realistically engaging in interactions. With these multiple constraints in place, it becomes relatively easy to identify vehicles that can genuinely interact.

For example, given the user input a vehicle suddenly accelerates and overtakes, the corresponding script includes the following steps: first, generating a trajectory for the acceleration behavior (agent.accelerate()); then, developing a trajectory for the overtaking behavior (agent.overtake()); and finally, calling a utility function to save the trajectory, outputting the trajectories of the ego vehicle and other agents in an array format. Below is an example of a cut in the trajectory generation function. We initialise the starting point coordinates and set a safety distance to ensure sufficient spacing between the agent and other agents. Additionally, we introduce random disturbances to the agent's speed and orientation in our simulation. By incorporating such randomness, we aim to closely approximate the uncertainties and variations typically encountered in practical driving environments. This approach helps to model better the unpredictable nature of traffic dynamics, sensor noise, and other real-world challenges, thereby improving the robustness of our system in dealing with such conditions.

```python
def cut_in(obj_trajs=None, obj_vels=None, safe_dis=10, is_ego=False):
    '''
    Generate a trajectory for an agent cutting in.
    Parameters:
    - obj_trajs: Trajectory of the target object.
    - obj_vels: Velocities of the target object.
    - safe_dis: Safe distance to maintain while cutting in.
    - is_ego: Whether the agent is the ego vehicle or not.

    Returns:
    - agent_trajs: The trajectory of the agent cutting in.
    - agent_vels: The velocities of the agent.
    '''
    # Initialize agent's trajectory and velocity arrays
    agent_traj = np.zeros((1, NUM_POINT, 3))
    agent_vel = np.zeros((1, NUM_POINT, 1))

    # Set initial position based on whether it's the ego or another
                                        vehicle
    if not is_ego:
        init_position = np.random.rand(2)
        init_position[0] = utils.get_random_value(init_position[0], [0,
                                                    10])
        sign_factor = 1 if random.random() > 0.5 else -1
        init_position[1] = utils.get_random_value(init_position[1],
                                                    Y_RANGE) * sign_factor
        vertical_margin = init_position[1] * sign_factor
        agent_traj[0, 0, :2] = init_position
        target_y_position = 0
    else:
        vertical_margin = abs(obj_trajs[0, 0, 1])
        target_y_position = obj_trajs[0, 0, 1]

    # Initialize agent's velocity
    velocity = utils.get_random_value(np.random.rand(), V_RANGE)
    agent_vel[0, 0] = velocity
    heading_angle = utils.get_random_value(np.random.rand(),
                                            FORWARD_RANGE)

    # Calculate initial velocity components
```

```
1188        velocity_x = velocity * np.cos(heading_angle)
1189        velocity_y = velocity * np.sin(heading_angle)
1190
1191        is_cutting_in = True
1192
1193        # Generate the cutting-in trajectory
1194        for t in range(1, NUM_POINT):
1195            agent_traj[0, t, 0] = agent_traj[0, t - 1, 0]  + \
1196                                  velocity_x * T_INTER
1197            agent_traj[0, t, 1] = agent_traj[0, t - 1, 1]  +  \
1198                                  velocity_y * T_INTER
1199
1200            # Check if the agent reaches the front of the target vehicle
1201            if (agent_traj[0, t - 1, 1] - vertical_margin) * \
1202                (agent_traj[0, t, 1] - vertical_margin) <= 0:
1203                is_cutting_in = False
1204
1205            # Approach the target vehicle
1206            if abs(agent_traj[0, t, 1] - target_y_position) \
1207                > vertical_margin / 2 and is_cutting_in:
1208                yaw_limit = [-0.1, 0] if sign_factor == 1 else [0, 0.1]
1209                max_yaw = 0 if sign_factor == 1 else 20 * np.pi / 180
1210                min_yaw = -20 * np.pi / 180 if sign_factor == 1 else 0
1211            # Gradually return to forward direction
1212            elif abs(agent_traj[0, t, 1] - target_y_position) \
1213                   > 0.5 and is_cutting_in:
1214                yaw_limit = [0, 0.1] if sign_factor == 1 \
1215                            else [-0.1, 0]
1216                max_yaw = -10 * np.pi / 180 if sign_factor == 1 \
1217                        else 20 * np.pi / 180
1218                min_yaw = -20 * np.pi / 180 if sign_factor == 1 \
1219                        else 10 * np.pi / 180
1220            # Move forward
1221            else:
1222                yaw_limit = [-0.3 * np.pi / 180, 0] if heading_angle >= 0 \
1223                            else [0, 0.3 * np.pi / 180]
1224                max_yaw = 1 * np.pi / 180
1225                min_yaw = -1 * np.pi / 180
1226
1227            # Fine-tune the direction to avoid collision
1228            if obj_trajs is not None:
1229                yaw_limit = utils.update_yaw(heading_angle, safe_dis,\
1230                            agent_traj, obj_trajs)
1231
1232            heading_angle += utils.get_random_value(np.random.rand(), \
1233                                            yaw_limit)
1234            heading_angle = np.clip(heading_angle, min_yaw, max_yaw)
1235
1236            # Update velocity and ensure it surpasses the other vehicle to
1237                                            realize cutting in
1238            velocity += utils.get_random_value(np.random.rand(), [-2, 2])
1239            if obj_vels is not None:
1240                velocity_range = [obj_vels[0, t, 0] + 0.5, V_RANGE[1] + 0.5]
1241            else:
                    velocity_range = V_RANGE
                velocity = np.clip(velocity, velocity_range[0], velocity_range[1]
                                                )
                agent_vel[0, t] = velocity

                # Update velocity components
                velocity_x = velocity * np.cos(heading_angle)
                velocity_y = velocity * np.sin(heading_angle)

        return agent_traj, agent_vel
```

**Trajectory Check.** To ensure the validity and safety of the generated trajectories and to prevent the agent's trajectory from deviating outside the drivable area, we employ a projection-based approach. Specifically, we project all relevant elements, such as drivable regions, static objects, and dynamic objects, onto the Bird's Eye View (BEV) perspective using the annotations provided in the dataset. This transformation allows for a clear and accurate representation of the environment from an overhead perspective, facilitating the detection of potential issues with trajectory generation. Simultaneously, we project the generated trajectory onto the same BEV perspective, enabling a direct comparison between the trajectory and the map features. By comparing the trajectory with the projected drivable regions and static and dynamic objects in the BEV space, we can effectively prevent the trajectory from exceeding the boundaries of the drivable area or colliding with other objects. To illustrate this process, we present a simple example in which we utilise map information and annotations from the nuScenes dataset, alongside newly generated trajectory data, to assess whether the trajectory remains within the defined driving area and avoids potential collisions with other agents in the environment.

| Interesting Adversarial Scenarios | CR |
|---|---|
| 4000 | 0.132 |
| 8000 | 0.111 |
| 12000 | 0.097 |
| 16000 | 0.083 |
| 20000 | 0.079 |
| 24000 | **0.072** |
| 28000 | 0.073 |

Table 16: Relationship between interesting adversarial scenarios count and final performance in synthetic adversarial scenarios.

```python
def check_trajectory(agent_trajectory, sample_token, nusc):
    '''
    Check if the agent is within the driving area \
    and if it collides with other agents.
    Parameters:
    - agent_trajectory: (N, 2) numpy array of the trajectory (x, y
                                    coordinates).
    - sample_token: The token of the sample in the nuScenes dataset.
    - nusc: The nuScenes dataset object.

    Returns:
    - A message indicating whether the trajectory is valid and if there
                                    are collisions.
    '''
    # Step 1: Get map information (roads, lanes, driving area)
    map_info = nusc.get_map_info()

    # Step 2: Check if the agent's trajectory is within the driving area
    def check_in_driving_area(trajectory, map_info):
        return np.all(trajectory[:, 0] >= 0) and \
               np.all(trajectory[:, 1] >= 0)

    if not check_in_driving_area(agent_trajectory, map_info):
        return "Agent's trajectory is outside the allowed driving area."

    # Step 3: Get other agents' trajectories for the same sample
    other_agents_trajectories = nusc.get_other_agents_trajectories(
                                    sample_token)

    # Step 4: Project the agent's trajectory and other agents'
                                    trajectories to BEV

    agent_bev_trajectory = utils.project_bev(agent_trajectory)
    other_agents_bev_trajectories = [utils.project_bev(trajectory) for
                                    trajectory in
                                    other_agents_trajectories]

    # Step 5: Check for potential collisions with other agents
    def check_collision(agent_trajectory, other_agents_trajectories):
        for other_trajectory in other_agents_trajectories:
            distance = np.linalg.norm(agent_trajectory - \
                       other_trajectory, axis=1)
```

```
        if np.any(distance < 1.0):
            return True
    return False

if check_collision(agent_bev_trajectory,
                        other_agents_bev_trajectories):
    return "Collision detected with another agent."

# If no collision and within driving area
return "Trajectory is within the driving area and does not collide
                    with other agents."
```

**Analysis of the DAA.** We conducted a more in-depth analysis of the DAA module, specifically focusing on the effect of the number of extreme case samples on final performance. In this experiment, we kept the number of non-corner-case scenarios constant while systematically varying the number of corner-case scenarios. As shown in Tables 16 and 17, the quantity of corner cases has a significant impact on the final performance, particularly the collision rate. Increasing the number of corner cases leads to a noticeable reduction in the collision rate. However, this improvement in safety comes at the cost of potential instability in the DR (Driving Risk) metric. The presence of a larger number of corner cases tends to steer

| Interesting Adversarial Scenarios | CR↓ | DR↓ |
|---|---|---|
| 4000 | 0.086 | 0.048 |
| 8000 | 0.084 | 0.043 |
| 12000 | 0.075 | 0.039 |
| 16000 | 0.063 | **0.035** |
| 20000 | 0.061 | 0.039 |
| 24000 | 0.059 | 0.041 |
| 28000 | **0.057** | 0.042 |

Table 17: Relationship between interesting adversarial scenarios count and final performance in the nominal real driving domain.

the policy toward more conservative driving behavior. While this shift reduces the occurrence of crashes, it may also induce hesitation in situations that require acceleration, potentially causing the system to fail to respond adequately to critical situations. This cautious behavior can ultimately result in yawing or other forms of instability under certain conditions.

## F    DETAILS OF EDITED SCENES

We design a series of corner cases inspired by (Dosovitskiy et al., 2017; Jia et al., 2024). As shown in Fig. 13 to 18, we present a diverse set of corner case scenarios, including BlockedIntersection, DynamicCutIn, OppositeLaneIntrusion, ParkingCutIn, HardBrake, and HazardAtSideLane. Then, we detail several corner cases from our edited scenes.

**BlockedIntersection.** While performing a maneuver, the ego vehicle encounters a stationary vehicle blocking the road and must take evasive action or apply the brakes to avoid a collision.

**DynamicCutIn.** The ego vehicle is required to slow down or brake in order to create a safe gap, allowing a vehicle from the adjacent lane to cut in front. This maneuver involves adjusting the ego vehicles speed to accommodate the other vehicle's movement, ensuring a smooth and safe transition without causing any disruption to the surrounding traffic flow or posing a risk of collision. The ego vehicle must carefully monitor its surroundings and adjust its position accordingly to maintain safe distances from both the vehicle cutting in and the vehicles around it.

**OppositeLaneIntrusion.** While driving straight, the ego vehicle encounters a situation where a vehicle from the opposite lane unexpectedly encroaches into its lane, creating an immediate risk of collision. In response to this threat, the ego vehicle is compelled to either brake rapidly to reduce speed or steer to the right to avoid the oncoming vehicle, all while carefully considering the surrounding traffic conditions and maintaining control of the vehicle to ensure a safe maneuver.

**ParkingCutIn.** The ego vehicle must reduce its speed or apply the brakes to allow a parked vehicle exiting a parallel parking space to merge into the lane ahead.

**HardBrake.** The leading vehicle decelerates abruptly, creating an unexpected situation where the ego vehicle must react quickly to avoid a collision. As a result, the ego vehicle is required to perform

an emergency stop, applying maximum braking force to rapidly reduce speed and prevent a rear-end collision, while carefully managing the braking distance and ensuring the vehicle remains under control throughout the maneuver.

**HazardAtSideLane.** The ego vehicle encounters a slow-moving obstacle partially blocking the lane and needs to maneuver into a lane of traffic moving in the same direction to bypass it. Unlike ParkedObstacle, this scenario places greater emphasis on merging into another lane.

**ParkedObstacle.** The ego vehicle encounters a parked car that is obstructing part of the lane, creating a situation where it cannot continue along its current path. To avoid the obstruction, the ego vehicle must maneuver by changing lanes into the moving traffic flowing in the same direction. This requires careful judgment and coordination, as the vehicle must assess the speed and proximity of nearby vehicles, ensuring it can safely execute the lane change without causing disruption or risk of collision.

**MergeIntoSlowTraffic.** In this scenario, the ego vehicle is required to merge into a slow-moving traffic flow, navigating through the congested lanes while ensuring safe integration with vehicles moving at a lower speed. The vehicle must carefully assess the surrounding traffic conditions, adjust its speed appropriately, and select an optimal gap to merge.

**Construction.** The ego vehicle encounters a construction zone blocking part of the lane and must shift into traffic moving in the same direction to navigate around it. Comparison to ParkedObstacle, the construction area occupies more of the lane's width, and the ego vehicle must temporarily divert from its planned route to bypass it.

**OppositeLaneRightTurn.** While preparing to make a right turn, the ego vehicle faces a situation where a vehicle from the opposite lane unexpectedly enters its lane, forcing the ego vehicle to take evasive action. In order to avoid a potential collision, the ego vehicle is compelled to quickly maneuver to the right, adjusting its position to maintain a safe distance from the incoming vehicle, all while preparing to complete the turn safely.

**BlindIntersectionCrossing.** The ego vehicle approaches an intersection where its view is obstructed by stationary objects or other vehicles, making it difficult to assess the traffic conditions and potential hazards ahead. As a result, the vehicle is required to slow down or come to a complete stop, proceeding cautiously only after confirming that the intersection is clear and it is safe to continue. This careful approach ensures that the vehicle avoids potential collisions or other risks that could arise from the limited visibility.

**WrongWayVehicle.** A vehicle traveling in the wrong direction rapidly approaches the ego vehicles lane, heading directly towards it in a head-on manner. Faced with this sudden and dangerous situation, the ego vehicle is forced to make a swift decision, either by immediately applying the brakes to reduce speed or by steering away to avoid a potential collision, all while considering the surrounding traffic and road conditions to ensure a safe maneuver.

**LaneChangeConflict.** Both the ego vehicle and a vehicle in an adjacent lane simultaneously attempt to merge into the same lane, requiring the ego vehicle to yield or abort the lane change.

**ParkedObstacleTwoWays.** The "TwoWays" version of ParkedObstacle. The ego vehicle faces a parked car blocking the lane and must change lanes into traffic moving in the opposite direction to avoid it.

**ConstructionTwoWays.** The "TwoWays" version of Construction. The ego vehicle encounters a construction area blocking the lane in both directions and must change lanes into traffic flowing in the opposite direction to avoid it. Compared to Construction, this obstruction occupies a wider lane area, requiring the ego vehicle to navigate through traffic in both directions.

**HazardAtSideLaneTwoWays.** The ego vehicle encounters a slow-moving obstacle that is partially blocking its lane, and in order to avoid a collision, it is faced with the difficult decision of either applying the brakes to slow down or executing a maneuver to steer into the oncoming traffic in the opposite direction.

# G  ADDITIONAL EXPERIMENTAL RESULTS.

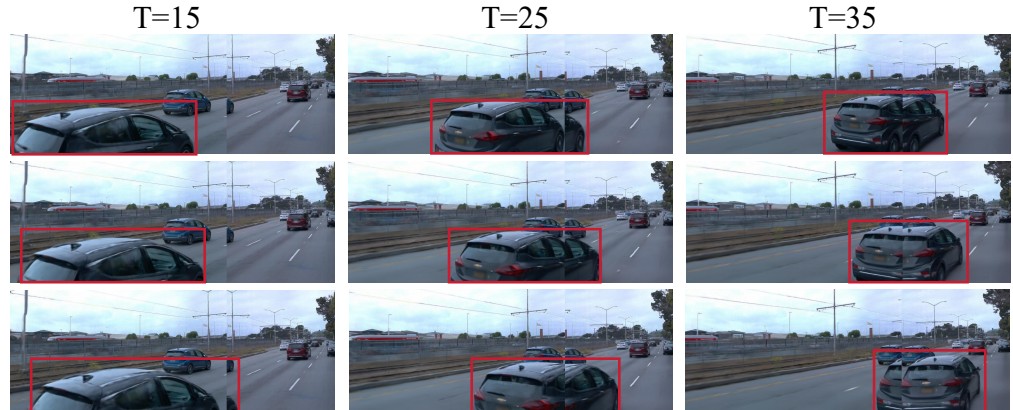

Figure 11: Various trajectories generated by the Dynamic Adversary Agent (DAA) for the vehicle in the red box at the same timestep.

| Method | Lane Shift @ 3m | | | Lane Shift @ 6m | | | Lane Change | | |
|--------|-----------|-----------|-------|-----------|-----------|--------|-----------|-----------|-------|
| | NTA-IoU ↑ | NTL-IoU ↑ | FID ↓ | NTA-IoU ↑ | NTL-IoU ↑ | FID ↓ | NTA-IoU ↑ | NTL-IoU ↑ | FID ↓ |
| w/o Video Diffusion Prior | 0.477 | 53.23 | 128.90 | 0.353 | 47.14 | 224.62 | 0.486 | 55.34 | 89.83 |
| w/ Video Diffusion Prior | **0.543** | **54.84** | **90.12** | **0.454** | **52.92** | **148.23** | **0.552** | **56.12** | **72.86** |

Table 18: Impact of diffusion prior within ReconSimulator on novel view rendering quality across lane shift and lane change on the Waymo (Sun et al., 2020).

**ReconSimulator.** We also perform ablation experiments on ReconSimulator to assess the impact of the diffusion prior (Ni et al., 2024; Zhao et al., 2024b; Wang et al., 2023) on novel view rendering quality, using the Waymo dataset (Sun et al., 2020), as shown in Tab. 18. Specifically, for the lane change, integrating the video

| Method | CR↓ | DCR↓ | SCR↓ |
|--------|-----|------|------|
| RAD (Gao et al., 2025) | 0.286 | 0.193 | 0.093 |
| *ReconDreamer-RL* | **0.147** | **0.100** | **0.047** |

Table 19: Comparison of collision metrics on the Waymo (Sun et al., 2020) dataset.

diffusion prior significantly improves the NTA-IoU by 13.58%, NTL-IoU by 1.41%, and reduces the FID score by 18.89%. These improvements clearly demonstrate that incorporating the video diffusion prior allows the rendered environment to provide better support for reinforcement learning tasks by enhancing the visual consistency and realism. Moreover, these results underline the strong generalization capability of ReconSimulator's appearance modeling across diverse datasets, highlighting its robustness and effectiveness across datasets.

**Quantitative Results on Waymo.** As shown in Tab. 19, we quantitatively compare the performance of *ReconDreamer-RL* and RAD (Gao et al., 2025) on the Waymo dataset (Sun et al., 2020). Specifically, *ReconDreamer-RL* achieves a 2× reduction in the Collision Ratio compared to RAD (Gao et al., 2025), demonstrating superior safety performance and enhanced robustness in complex driving scenarios.

**Qualitative Results on Waymo.** As shown in Fig. 12, we compare *ReconDreamer-RL* and RAD (Gao et al., 2025) under two corner cases: HardBrake and DynamicCutIn. *ReconDreamer-RL* successfully avoids collisions by properly adjusting its speed, benefiting from a more realistic simulator enabled by diffusion-based scene reconstruction and further enhanced through data augmentation.

**Visualization of DAA.** As mentioned in the main text, in the reinforcement learning stage, the Dynamic Adversary Agent (DAA) has a certain probability of fine-tuning the trajectories in the edited scenes (e.g., adjusting the target vehicle's speed) instead of directly reusing those generated during the imitation stage. In Fig. 11, we present different trajectories generated by DAA at the

*ReconDreamer-RL*                    RAD

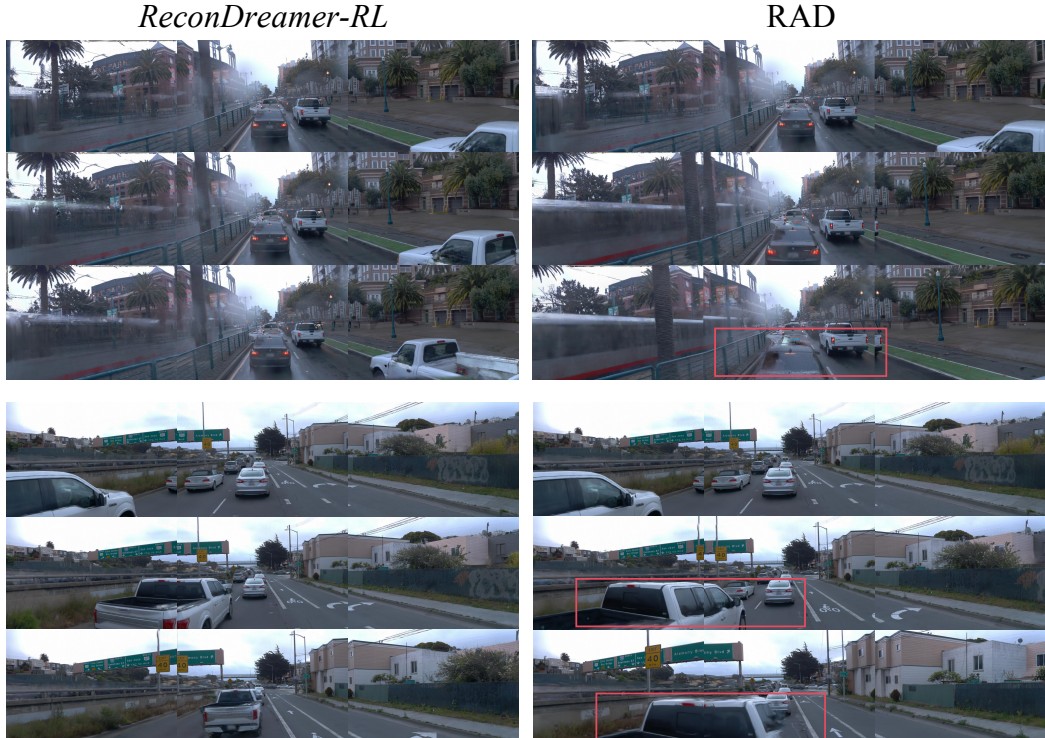

Figure 12: Comparison of different methods in challenging corner cases, with collisions highlighted by red boxes.

same timestep. Since the method directly modifies the speed based on the original trajectory, it does not affect the overall timing of the pipeline.

## H    THE USE OF LARGE LANGUAGE MODELS

In this work, large language models (LLMs) are employed solely for textual refinement purposes. Specifically, they are used to polish the manuscript in order to improve readability, clarity, and consistency, without influencing the methodology or experimental results. This ensures that the role of LLMs remains confined to language editing, while all technical contributions and findings are produced independently.

## I    SUPPLEMENTARY VIDEO

We provide a video that showcases additional comparisons between *ReconDreamer-RL* and RAD (Gao et al., 2025) across diverse datasets. For further details, please refer to the file located at video/comparison.mp4.

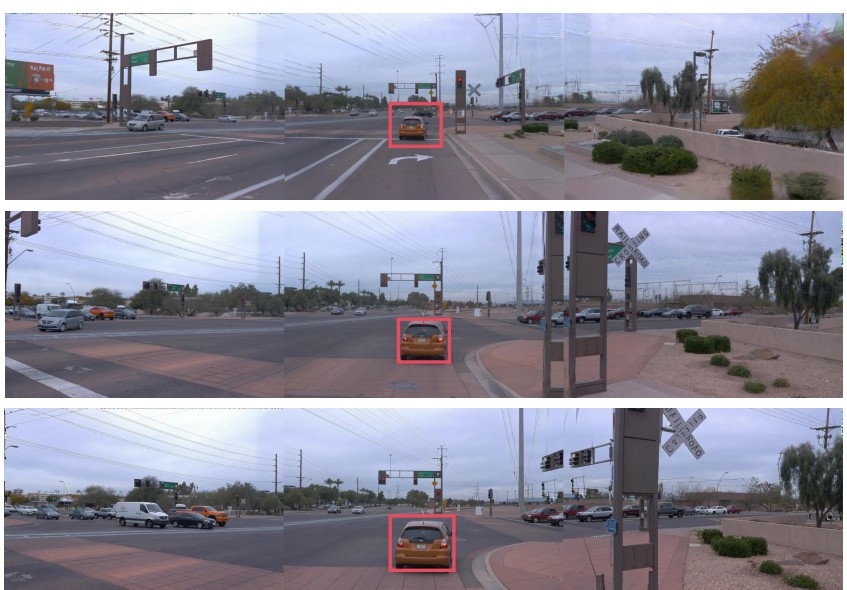

Figure 13: Example scenario of BlockedIntersection. The vehicle in the red box is stationary and blocks the intersection when the ego vehicle intends to turn. The ego vehicle must brake or change lanes to the left to avoid a potential collision.

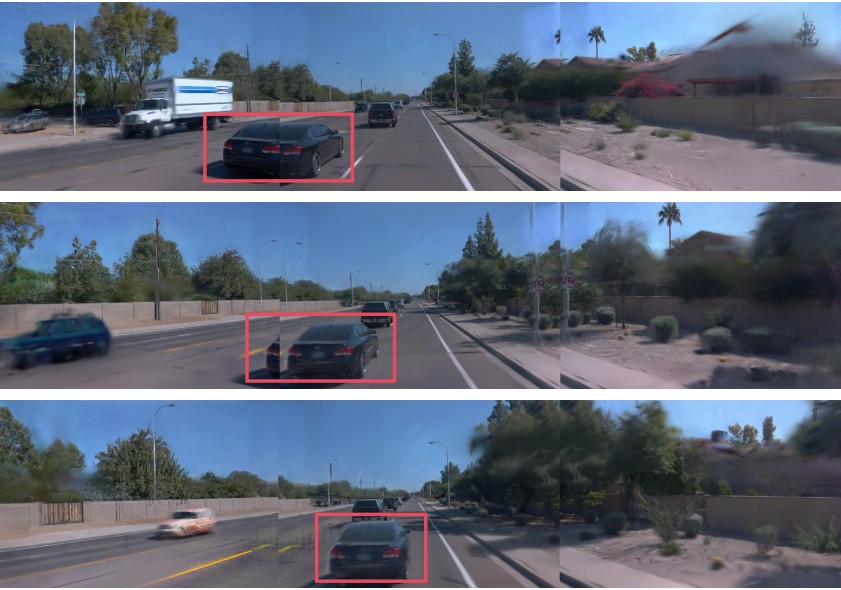

Figure 14: Example scenario of DynamicCutIn. The vehicle in the red box is attempting to cut in from an adjacent lane while the ego vehicle is moving straight. The ego vehicle needs to slow down or brake to yield.

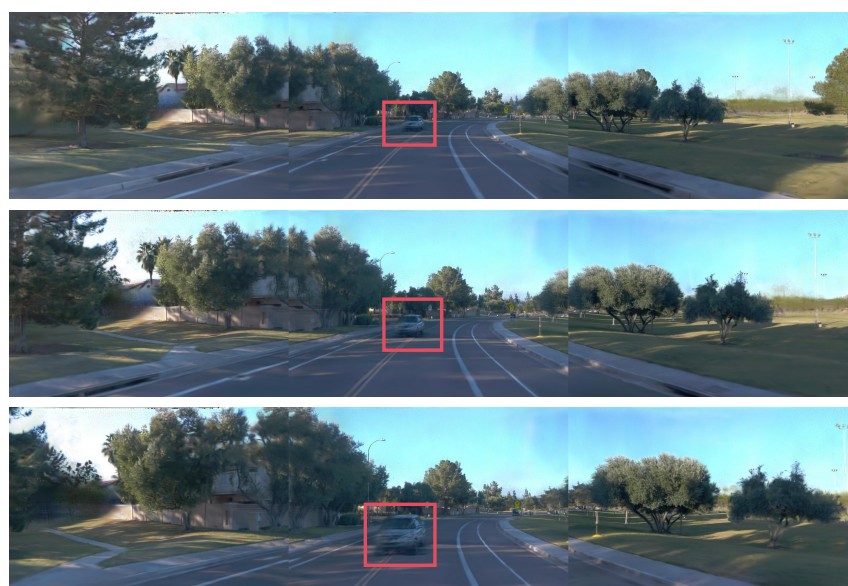

Figure 15: Example scenario of OppositeLaneIntrusion. The vehicle in the red box is attempting to encroach into the ego vehicle's lane from the opposite direction. The ego vehicle must slow down or steer away to avoid a collision.

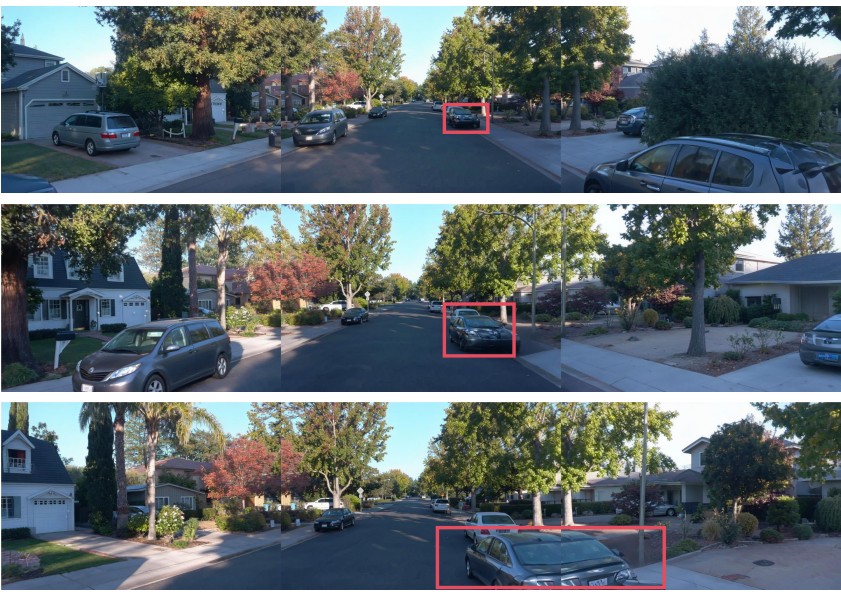

Figure 16: Example scenario of ParkingCutIn. The vehicle in the red box is attempting to exit a parallel parking space and merge into the lane ahead. The ego vehicle must slow down or stop to avoid a collision.

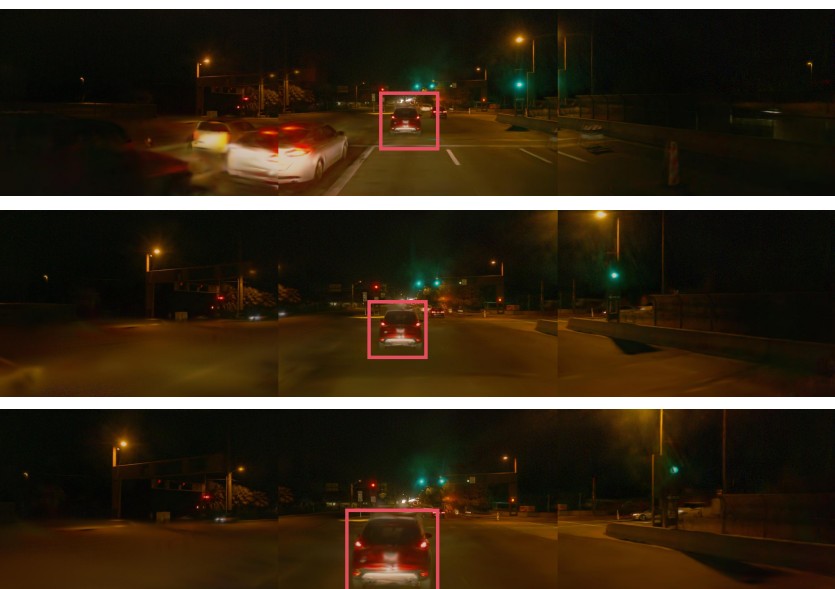

Figure 17: Example scenario of HardBrake. The vehicle in the red box is decelerating abruptly. The ego vehicle needs to change lanes to avoid a rear-end collision.

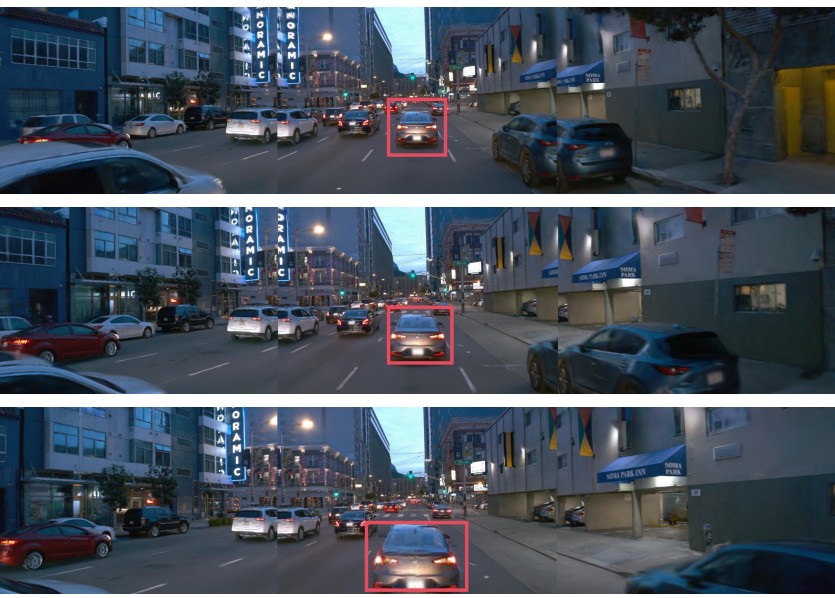

Figure 18: Example scenario of HazardAtSideLane. The vehicle in the red box is slowly moving and partially occupying the ego vehicle's lane. The ego vehicle needs to change lanes to the left and merge into the traffic moving in the same direction.

