# OpenReview forum: "ReconDreamer-RL: Enhancing Reinforcement Learning via Diffusion-based Scene Reconstruction"
_ICLR.cc/2026/Conference — Submitted to ICLR 2026_

### Official Review · Reviewer_d1uV · 2025-10-29

**Soundness:** 3
**Presentation:** 3
**Contribution:** 3
**Rating:** 6
**Confidence:** 3

**Summary:**

* This paper addresses the sim2real gap in training e2e autonomous driving models with RL.
* It proposes ReconDreamer-RL, a framework that integrates video diffusion priors to create more realistic and robust simulation environments for RL training.
* ReconSimulator: A simulator that uses 3DGS for reconstruction, a video diffusion prior for realistic appearance modeling of novel views, and a kinematic model.
* Dynamic Adversary Agent (DAA): Generates challenging traffic scenarios by controlling surrounding vehicles.
* Cousin Trajectory Generator (CTG): Enriches training data by synthesizing diverse ego trajectories to reduce simple straight driving.
* Training happens in two stages: 1) IL for policy initialization using dataset from CTG / DAA. 2) Closed-loop RL stage where policy interacts with ReconSimulator and adversarial DAA.
* Experiments demonstrate that this framework significantly outperforms baseline IL methods and achieves a new SOTA in RL + 3DGS.

**Strengths:**

* This paper focuses on a highly relevant research direction for AVs.
* The problem of reducing the sim2real gap and training e2e models with RL is addressed holistically with improvements in the simulator’s fidelity (3DGS with video diffusion prior), focusing on long-tail problems (DAA), and leverage diverse ego trajectories (CTG).
* Overall, the results demonstrate the superior performance of ReconDreamer-RL, e.g. a 3x reduction in collisions rates over baselines. And the method is validated on both nuScenes and the Waymo Open Dataset.
* The ablations also clearly demonstrate the value that each component brings.

**Weaknesses:**

* Lack of trajectory generation function implementation details. The authors write e.g. text-to-trajectory can be used but it seems that this paper lacks any additional details about the implementation used here. Given the importance of the DAA in the ablations, more details should be provided here.
* Unclear computational cost of the iterative scene reconstruction. You provide information about fast rendering speeds (125 fps) but not about the scene reconstruction, which seems particularly expensive. Can you provide more details here, e.g. how many steps are usually needed?

**Questions:**

* How does the collision avoidance in the CTG work and how does CTG interact with the more adversarial behaviors created from the DAA?

---

> ### Author Response · Authors · 2025-11-24
> **Rebuttal by Authors (continued)**
>
> **Dear Reviewer d1uV,**
>
> We appreciate your feedback and thoughtful comments. In response to the points you've raised, we have provided the following clarifications:
>
>
> > **[W1] Lack of trajectory generation function implementation details.**
>
>  [A1] Thank you for your valuable suggestions to provide more details on the implementation of the trajectory generation function. In fact, we use a text-to-trajectory model as the DAA module's function. Specifically, we first construct a comprehensive trajectory-generation function library that covers a wide range of agent behaviors, such as steering, acceleration, and braking. Leveraging this library, we curate a paired text-to-trajectory dataset to fine-tune a large language model (LLM), enabling it to automatically generate diverse agent trajectories from natural language descriptions. We provide more details in Appendix E, including the full pipeline used to generate trajectories and example code for the trajectory-generation function library.
>
>
>
> > **[W2] Unclear computational cost of the iterative scene reconstruction.**
>
>
> [A2] For iterative scene reconstruction, the integration of the diffusion prior introduces minimal overhead as it only requires inference without any additional training. Moreover, our method adds approximately 10 percent more optimization steps, around 4000 steps in total, which remains fully manageable in terms of computational cost. Compared to RAD-3DGS, our method incurs an additional runtime of only about 0.3 hours per scene, as shown in the table below. We have added this information to Appendix C.
>
> Meanwhile, we compare different strategies for the ReconSimulator to reconstruct a scene. First, we train a DriveRestorer model on the Waymo dataset and apply it directly to the ReconSimulator for reconstructing the nuScenes dataset, assessing both computational overhead and generalization performance. As shown in the table below, when the ReconSimulator utilizes the DriveRestorer model trained on the Waymo dataset for reconstruction, it still significantly improves reconstruction on nuScenes, demonstrating strong cross-dataset generalization. This generalization substantially reduces the cost of integrating the diffusion prior. Moreover, fine-tuning DriveRestorer on a target dataset is relatively inexpensive, requiring only 3 hours, whereas training from scratch takes 12 hours, which remains reasonable. Additionally, this overhead is incurred only once for reconstructing all the scenes in the target dataset.
>
>
>
> |    &nbsp;&nbsp;&nbsp; &nbsp;&nbsp;&nbsp;&nbsp;&nbsp;&nbsp;&nbsp;&nbsp; &nbsp;&nbsp;&nbsp;&nbsp;&nbsp;&nbsp;   Model     | NTA-IoU ↑ | NTL-IoU ↑ | FID ↓   | Extra Diffusion Training Time (h) | Single Reconstruction Time (h) |
> | :--------------------------------: | :-----------: | :-----------: | :---------: | :--------------------------------------: | :-------------------------------------: |
> | RAD-3DGS                       | 0.219     | 48.72     | 149.50  | --                                   | 1.5                                 |
> | ReconSimulator w/o finetune    | 0.287     | 49.75     | 144.31  | --                                   | 1.8                                 |
> | ReconSimulator w/ finetune     | 0.317     | **49.84** | 127.10  | 3.0                                  | 1.8                                 |
> | ReconSimulator re-trained      | **0.325** | 49.72     | **125.43** | 12.0                               | 1.8                                 |
>
> **To be continued in next reply.**

---

> ### Author Response · Authors · 2025-11-24
> **Rebuttal by Authors (continued)**
>
> > **[Q1]How does the collision avoidance in the CTG work, and how does CTG interact with the more adversarial behaviors created from the DAA?**
>
> [A3] To ensure collision-free generated trajectories and prevent the agent’s trajectory from deviating outside the drivable area, we project all drivable regions, static objects, and dynamic agents onto the Bird's Eye View (BEV) perspective based on dataset annotations. The generated trajectory is also projected into the same BEV space. Subsequently, within this BEV representation, we compute the distances between the trajectory and each agent as well as the map boundaries to explicitly verify the absence of collisions and boundary violations.
>
> The Dynamic Adversary Agent (DAA) module generates corner-case traffic scenarios by adjusting the trajectories of surrounding vehicles relative to the ego vehicle. Meanwhile, the Cousin Trajectory Generator (CTG) addresses the issue of biased training data that typically favors simple, straight-line movements. It generates more complex and varied ego trajectories, ensuring a broader range of driving scenarios in the training process. These two modules do not directly interact at the moment; rather, they independently generate data that is used together for policy training. In the future, we plan to explore the potential interaction between these modules to further enhance performance and achieve even more effective results.
>
>
> ---
>
> We would like to express our sincere gratitude to Reviewer d1uV for your detailed and constructive feedback. We have made every effort to address all of your concerns and hope that our responses provide the necessary clarifications. Should you have any additional questions, please do not hesitate to contact us! We remain fully committed to engaging in the discussion during the rebuttal period and look forward to your further insights!

---

> > ### Comment · Reviewer_d1uV · 2025-11-25
> >
> > Thank you for the detailed response and the details on the trajectory generation function!

---

### Official Review · Reviewer_mSxh · 2025-10-30

**Soundness:** 3
**Presentation:** 3
**Contribution:** 3
**Rating:** 6
**Confidence:** 3

**Summary:**

This paper proposes ReconDreamer‑RL, a framework for end‑to‑end autonomous‑driving policy training that couples (i) a reconstruction‑based simulator (ReconSimulator) enhanced by a video‑diffusion prior for appearance modeling with (ii) kinematic physical modeling of dynamic agents, and augments training with two data‑generation modules: a Dynamic Adversary Agent (DAA) that synthesizes corner‑case interactions and a Cousin Trajectory Generator (CTG) that diversifies ego trajectories. Training proceeds in two stages: imitation learning (IL) for initialization and reinforcement learning (RL) in closed loop. Experiments (nuScenes reconstruction as the main setting; Waymo for generalization) report large reductions in collision rate relative to IL baselines and sizable gains over the best prior RL baseline (RAD), while maintaining real‑time rendering speed.

**Strengths:**

1. The paper shows strong empirical gains on collision metrics across standard and corner‑case evaluations; large gap over both IL baselines and RAD.
2. RL‑friendly simulator: 3DGS‑based with diffusion priors while retaining real‑time speed (Table 2). This addresses a common bottleneck in closed‑loop training.

**Weaknesses:**

1. The policy is evaluated in the same simulator family that was fine‑tuned using diffusion priors, and many corner cases are generated by the paper’s own procedures (DAA, CTG). The real sim‑to‑real implications remain untested; no closed‑loop evaluation outside the authors’ reconstructions (e.g., Bench2Drive/nuPlan/CARLA‑v2 evaluations or any real‑vehicle trials).
2. Metrics are limited for driving quality. Collisions are critical, but comfort (jerk/accel), traffic‑rule compliance, and route completion are absent. The paper relies on PDR/HDR (Table 1) for trajectory adherence on unedited clips and on collision‑only metrics for edited scenes (Appx. A.2). This may overemphasize safety at the expense of smoothness and legality.

**Questions:**

Can you report RAD trained with the same DAA‑generated scenarios and/or CTG‑augmented data (keeping rendering identical), and vice‑versa, to isolate where the gains come from?

---

> ### Author Response · Authors · 2025-11-24
> **Rebuttal by Authors**
>
> **Dear Reviewer mSxh,**
>
> Thank you for your recognition and constructive feedback. In light of your concerns, we present our itemized responses to your comments below.
>
> ---
> > **[W1] ...no closed‑loop evaluation outside the authors’ reconstructions (e.g., Bench2Drive/nuPlan/CARLA‑v2 evaluations or any real‑vehicle trials).**
>
> [A1]  For the table below, we conducted closed-loop simulations within the NuPlan framework, and the experimental results demonstrate that testing closed-loop evaluations, beyond the reconstructions of ReconDreamer-RL, still exhibit strong superiority.
>
> |       &nbsp;&nbsp;&nbsp;&nbsp;&nbsp;&nbsp;&nbsp;&nbsp;&nbsp;Method                | NC ↑  | DAC ↑  | TTC ↑  | Comf ↑  | EP ↑  | PDMS ↑ |
> | :-----------: | :-----------: | :-----------: | :-----------: | :-----------: | :-----------: | :-----------: |
> | VADv2                 | 97.1  | 96.4   | 94.5   | **100**     | 80.9  | 89.3   |
> | RAD                   | 97.6  | 97.1   | 94.9   | **100**     | 81.6  | 88.9   |
> | ReconDreamer-RL       | **98.0**  | **97.7**   | **95.8**   | **100**     | **81.9**  | **89.4**   |
>
>
>
> > **[W2]...Collisions are critical, but comfort (jerk/accel), traffic‑rule compliance, and route completion are absent...**
>
> [A2] We conduct additional experiments to evaluate further metrics, including comfort (jerk/acceleration). As shown in the Table 1, we compare the comfort analysis in unedited 3DGS environments. We observe that *ReconDreamer-RL* performs similarly to RAD on Long. Jerk and Lat. Jerk metrics, and significantly outperforms algorithms that rely solely on imitation learning. This is because *ReconDreamer-RL* alternates between imitation and reinforcement learning during training, as detailed in Appendix C. This approach allows *ReconDreamer-RL* not only to improve safety but also to better align with human driving behavior.
> Additionally, we compare comfort metrics in cut-in scenarios, evaluating only scenes with no collisions across all algorithms. As shown in Table 2, ReconDreamer-RL consistently achieves the highest performance across these scenarios. This is attributed to its ability to learn more effective collision avoidance strategies from a diverse and extensive set of corner cases. In contrast, other methods that lack exposure to such challenging data exhibit instability when confronted with similar situations. We have incorporated this discussion into Section 4.3 of the main text.
>
>
> **Table 1: Comfort analysis in unedited 3DGS environments.**
>
> | &nbsp;&nbsp;&nbsp;&nbsp;&nbsp;&nbsp;Method  | Long. Jerk↓ | Lat. Jerk↓ |
> | :--------------------------------: | :-----------: | :-----------: |
> | VAD    | 5.645       | 0.597      |
> | GenAD  | 12.490      | 0.432      |
> | VADv2  | 8.124       | 0.435      |
> | RAD    | **4.237** | 0.191      |
> | *ReconDreamer-RL* | 4.264   | **0.189** |
>
>
> **Table 2: Comfort analysis in cut-in scenarios.**
>
> | &nbsp;&nbsp;&nbsp;&nbsp;&nbsp;&nbsp;Method  | Long. Jerk↓ | Lat. Jerk↓ |
> | :--------------------------------: | :-----------: | :-----------: |
> | VAD    | 6.714       | 0.784      |
> | GenAD  | 14.216      | 0.548      |
> | VADv2  | 9.245       | 0.499      |
> | RAD    | 4.912       | 0.316      |
> | *ReconDreamer-RL* | **3.832** | **0.276** |

---

> ### Author Response · Authors · 2025-12-02
> **Rebuttal by Authors (continued)**
>
> > **[Q1] Can you report RAD trained with the same DAA‑generated scenarios and/or CTG‑augmented data (keeping rendering identical), and vice‑versa, to isolate where the gains come from?**
>
> [A3] Thank you for your suggestion to include additional ablation studies on ReconSimulator. In rows 2, 3, and 4 of the table below, we replace ReconSimulator with RAD-3DGS to generate DAA and CTG data for training the RAD policy, while keeping the rendering pipeline unchanged. Despite RAD-3DGS's lower rendering fidelity, performance still improves because the original training data contains almost no corner-case scenarios. Although the synthesized scenes have lower visual quality, they still provide useful behavioral diversity that enhances policy learning. In contrast, as shown in rows 6, 7, and 8 of the table, using ReconSimulator to synthesize CTG and DAA data yields substantially greater performance gains. We have incorporated this analysis into Section 4.3 of the main text.
>
> | ReconSimulator | DAA | CTG | CR↓    | DCR↓   | SCR↓   | DR↓    | PDR↓   | HDR↓   |
> |----------------|-----|-----|--------|--------|--------|--------|--------|--------|
> |                |     |     | 0.238  | 0.143  | 0.095  | 0.084  | 0.057  | 0.027  |
> |                | ✓   |     | 0.167  | 0.102  | 0.065  | 0.076  | 0.052  | 0.024  |
> |                |     | ✓   | 0.191  | 0.121  | 0.070  | 0.068  | 0.046  | 0.022  |
> |                | ✓   | ✓   | 0.142  | 0.082  | 0.060  | 0.063  | 0.043  | 0.020  |
> | ✓              |     |     | 0.172  | 0.103  | 0.069  | 0.073  | 0.053  | 0.020  |
> | ✓              | ✓   |     | 0.117  | 0.069  | 0.048  | 0.067  | 0.050  | 0.017  |
> | ✓              |     | ✓   | 0.143  | 0.086  | 0.057  | 0.053  | 0.040  | 0.013  |
> | ✓              | ✓   | ✓   | **0.077** | **0.048** | **0.029** | **0.040** | **0.027** | **0.013** |
>
>
> ---
> We truly appreciate your thoughtful and expert feedback, which has greatly strengthened and refined our work. If you have any additional questions or require further clarification, please don't hesitate to reach out. We would be more than happy to provide any further explanations.

---

### Official Review · Reviewer_BUDD · 2025-10-31

**Soundness:** 3
**Presentation:** 3
**Contribution:** 3
**Rating:** 6
**Confidence:** 3

**Summary:**

This paper presents ReconDreamer-RL, a framework designed to improve the training of end-to-end autonomous driving agents through reinforcement learning by addressing the critical simulation-to-reality (sim2real) gap. The authors identify a key limitation in existing scene reconstruction-based simulators: their inability to render high-quality sensor data for novel trajectories or rare corner cases not present in the original training data. To overcome this, the proposed method integrates video diffusion priors into the simulation process. The core of the framework is the ReconSimulator, which combines a diffusion-based appearance model with a kinematic physical model to reconstruct realistic driving scenarios. To further enhance the training process and cover a wider range of challenging situations, the paper introduces two additional components: the Dynamic Adversary Agent (DAA), which autonomously generates corner-case traffic scenarios by adjusting the trajectories of surrounding vehicles, and the Cousin Trajectory Generator (CTG), designed to mitigate the common bias in training data towards simple, straight-line movements.

**Strengths:**

1. The paper introduces a novel method of integrating video diffusion priors directly into the reinforcement learning loop, creating a dynamic simulator that effectively addresses the sim2real gap.

2. It proposes a well-structured solution with dedicated modules (DAA and CTG) to systematically generate adversarial corner cases and mitigate common training data biases.

3. The framework demonstrates a significant and quantifiable performance improvement, achieving a 5x reduction in collision ratio over strong imitation learning baselines, providing clear evidence of its effectiveness.

**Weaknesses:**

1. Lack of originality. Most techniques proposed in the paper have already been well studied in other works. Using video models to boost novel-trajcectory reconstruction performance is studied in many works such as Drivedreamer4d[1]; Decoupled static and dynamic scene representation is proposed in MTGS[2], OmniRe[3]; Adversary agent interaction is studied in a line of research featuring safty-critical interaction such as DiffScene[4].

2. Missing details in the dynamic adversary agent section. How exactly is the trajectory generated and how to perform sanity check for the generated trajectory?

[1] Drivedreamer4d: World models are effective data machines for 4d driving scene representation; Zhao, et.al.

[2] MTGS: Multi-Traversal Gaussian Splatting; Li, et.al

[3] OmniRe: Omni Urban Scene Reconstruction; Chen, et.al.

[4] DiffScene: Guided Diffusion Models for Safety-Critical Scenario Generation

**Questions:**

See Weaknesses.

---

> ### Author Response · Authors · 2025-11-24
> **Rebuttal by Authors**
>
> **Dear Reviewer BUDD,**
>
> We are grateful for your careful review and constructive feedback. Your insights have been instrumental in refining our manuscript. Our point-by-point responses are presented below.
>
> ---
> > **[W1] Lack of originality.**
>
> [A1]  Our main contribution lies in enhancing reinforcement learning through diffusion-based scene reconstruction. Additionally, we will open-source the entire reconstruction environment, including 3DGS reconstructed from various datasets. This resource can significantly aid in training and evaluating end-to-end autonomous driving algorithms, making a valuable contribution to the community. Moreover, we provide further details and analysis of the training strategy for reinforcement learning using 4D reconstruction with a diffusion prior in Appendix D, which can drive future progress in autonomous driving research. Additionally, we have expanded the related work section to highlight how our contributions differentiate from these approaches [1-4], as detailed in the main text.
>
> > **[W2] Missing details in the dynamic adversary agent section. How exactly is the trajectory generated, and how to perform a sanity check for the generated trajectory?**
>
> [A2] Thank you for your valuable suggestions to provide more details on the implementation of the Dynamic Adversary Agent. Specifically, for trajectory generation, we utilize a text-to-trajectory model within the DAA module. We first construct a comprehensive trajectory-generation function library that captures a wide range of agent behaviors, including steering, acceleration, and braking. Using this library, we curate a paired text-to-trajectory dataset to fine-tune a large language model (LLM), enabling it to automatically generate diverse agent trajectories from natural language descriptions.
>
> Regarding collision checks, we project all drivable regions, static objects, and dynamic agents into the BEV perspective using dataset annotations. The generated trajectory is also projected into the same BEV space. In this BEV representation, we compute the distances between the trajectory and each agent, as well as the map boundaries, to explicitly ensure the absence of collisions and boundary violations. Meanwhile, we have further introduced Appendix E to provide more detailed information regarding the DAA， including the complete pipeline used for trajectory generation, along with example code for the trajectory-generation function library.
>
>
>
> **Reference:**
>
> [1] Drivedreamer4d: World models are effective data machines for 4d driving scene representation; Zhao, et.al.
>
> [2] MTGS: Multi-Traversal Gaussian Splatting; Li, et.al
>
> [3] OmniRe: Omni Urban Scene Reconstruction; Chen, et.al.
>
> [4] DiffScene: Guided Diffusion Models for Safety-Critical Scenario Generation
>
> ---
>
> We sincerely value your insightful and professional feedback, which has significantly enhanced and improved our work. Should you have any further questions or need additional clarification, please feel free to contact us. We would be delighted to offer any further explanations.

---

### Official Review · Reviewer_AWCj · 2025-11-03

**Soundness:** 3
**Presentation:** 2
**Contribution:** 3
**Rating:** 4
**Confidence:** 5

**Summary:**

This work proposes ReconDreamer-RL, a two-stage framework that strengthens end-to-end autonomous-driving training by bridging imitation learning with reinforcement learning inside a photorealistic, editable simulator. The framework starts with ReconSimulato to fuse video-diffusion priors for high-quality novel-view rendering with a kinematic model for physically valid agent motion. On top of sensor simulator, a Dynamic Adversary Agent (DAA) auto-generates corner-case interactions (e.g., cut-ins) and a Cousin Trajectory Generator (CTG) diversifies ego behaviors to counter dataset bias. Together these components reduce the sim-to-real gap and improve closed-loop robustness, yielding large drops in collision metrics over IL/RL baselines while maintaining fast rendering suitable for RL training.

**Strengths:**

- **Significance and Scope.** The paper addresses an important and timely problem in autonomous driving—closed-loop, end-to-end learning—by coupling photorealistic simulation with reinforcement learning in a way that directly targets deployment-relevant failures.
- **Comprehensive Qualitative Evaluation.** The study presents rich qualitative evidence across multiple benchmarks and driving scenarios, helping to substantiate generality and offering interpretable insights through scenario visualizations and case studies.
- **Strong Empirical Gains with Targeted Guidance.** Leveraging the Dynamic Adversary Agent (DAA) and Cousin Trajectory Generation (CTG), ReconDreamer-RL achieves consistent improvements over RAD and other E2E baselines on key safety and efficiency metrics in closed-loop settings.
- **Well-Designed Ablations.** Ablation studies are carefully constructed to isolate the contributions of ReconSimulator, DAA, and CTG, demonstrating each component’s necessity and the synergy of the full system.

**Weaknesses:**

Despite the strengths mentioned above, I have some significant concerns on the technical contribution claimed by the paper, listed in this and the question section below:

- **Unclear presentation of the ReconSimulator**. The core technical contribution of this work comes from this ReconSimulator, however, the details of this simulator is unclear. For instance, How is the diffusion prior incorporated? From section 3.3, it seems that the ReconSimulator use a training checkpoint rendered by 3DGS and refined by DriveRestorer. What if we directly utilize 3DGS with decomposed foreground and background, just like what is done in HUGISM [1]?
- **Limited contribution in the RL algorithm**. The title of this paper is ReconDreamer-RL, yet the novelty and analysis on the RL algorithm is missing in the main text. The authors mention in the appendix that they use PPO algorithm following RAD, which seems to be based on RAD with some incremental modification in synthetic data on the sensor simulation side. See more in the question part.

> [1] Zhou, Hongyu, et al. "Hugsim: A real-time, photo-realistic and closed-loop simulator for autonomous driving." *arXiv 2024.

**Questions:**

As mentioned above, I believe there are lots of merits in the paper, while I also have concerns about the key technical contribution, since they are not well-presented in the current manuscript. I will consider adjusting my rating later if the weaknesses and questions are properly addressed later.

- Do the authors empirically observe any forgetting issues when training on the synthetic data, if tested back into the real-domain with nominal driving cases?
- In line 265, what is the $f(\cdot)$ specifically defined in the DAA module? What is the sample scale needed for DAA to get interesting adversaries. If we conduct random samples, would most of the samples remain boring without challenging behaviors?
- In Figure 6, the author mentions cousin-nuScenes created by CTG from the original nuScenes dataset. This is primarily used to cold-start the Rl algorithm. In the online rollout in the PPO, what is the data diversity, sample scale, as well as their impact to the final perfomance? An ablation study on the final performance vs sample complexity will be interesting.
- Is DAA and CTG used to only cold-start the training data for RL, or also incorporated during the RL process? If they are both used during online learning, how are they used?

---

> ### Author Response · Authors · 2025-11-24
> **Rebuttal by Authors**
>
> **Dear Reviewer AWCj,**
>
> Thank you for your insightful comments and thoughtful questions. They have contributed significantly to enhancing the quality of our work. Our responses are provided below.
>
> ---
> > **[W1] Unclear presentation of the ReconSimulator.**
>
> [A1]  Thank you for your valuable suggestions to provide more details on ReconSimulator. The process of integrating the diffusion prior is divided into two stages. In the first stage, we train the scene reconstruction model $G$ using the original data $V_{ori}$ for $k_1$ steps until the rendered quality meets an acceptable level from the perspective of $\mathcal{T}_{ori}$, such as PSNR > 28. The reconstruction loss is:
>
> $$
> Loss = \lambda_1 L_{RGB}^{ori} + \lambda_2 L_{Depth}^{ori} + \lambda_3 L_{SSIM}^{ori}.
> $$
>
> In the second stage, the trained model $G$ is used to render novel trajectory videos $\hat{V} _ {novel} = G(\mathcal{T} _ {novel})$, where $\mathcal{T}_{novel}$ is the novel trajectory. These rendered videos often exhibit ghost artifacts due to data sparsity. DriveRestorer is then introduced to fix these artifacts by freezing its network parameters. The inference process can be expressed as:
>
> $$
> V_{novel} = \mathcal{R}(\hat{V}_{novel}, c),
> $$
>
> where $\mathcal{R}$ is the DriveRestorer model, and $c$ represents the structural conditions corresponding to the degraded video $\hat{V }_ {novel}$. A mixed dataset $V$ is then constructed by combining the original trajectory video dataset
> $V_{ori}$ and the restored novel trajectory video dataset $V_{novel}$:
>
> $$
> V = 0.5 \cdot V_{ori} \cup 0.5 \cdot V_{novel}.
> $$
>
> This mixed dataset is used to further train the scene reconstruction model $G$ for $k_2$ steps. The further training enables $G$ to render high-quality images across a broader range of trajectories. To meet the requirements of reinforcement learning for free exploration of scenes, we gradually update $V_{novel}$. Specifically, during every $k_3$-th step of training $G$, a new trajectory $\mathcal{T} _ {novel}'$ is selected, and the reconstruction model $G$ renders new trajectory videos $\hat{V} _ {novel}'$, which are processed by DriveRestorer to obtain the restored novel trajectory videos $V_{novel}'$. These videos are then added to the $V_{novel}$ dataset. To ensure that the newly generated data provides additional priors for the reconstruction model, the updated dataset $V_{novel}$ is modified as:
>
> $$
> V_{novel} = (1 - w) \cdot V_{novel} \cup w \cdot V_{novel}',
> $$
>
> where $w$ is the sampling probability, in this stage, the loss function is:
>
> $$
> Loss = \lambda_1 L_{RGB}^{ori} + \lambda_2 L_{Depth}^{ori} + \lambda_3 L_{SSIM}^{ori} + \lambda_4 L_{RGB}^{novel} + \lambda_5 L_{SSIM}^{novel},
> $$
>
> where $L_{RGB}^{ori}$ is the RGB reconstruction loss for the original dataset, $L_{Depth}^{ori}$ is the depth loss, $L_{SSIM}^{ori}$ is the structural similarity loss for the original dataset, $L_{RGB}^{novel}$ is the RGB reconstruction loss for the novel trajectory dataset, and $L_{SSIM}^{novel}$ is the structural similarity loss for the novel trajectory dataset. Notably, the synthetic dataset does not use depth as supervision. We have added this content in Appendix C. Additionally, HUGISM, Street Gaussians, and our ReconSimulator all utilize foreground and background decomposition. However, unlike HUGISM and RAD-3DGS, which only supervise views along the ego-vehicle trajectory, our method allows for exploration of viewpoints beyond this path. When reinforcement learning explores viewpoints outside the ego-vehicle trajectory, the rendering quality of HUGISM and RAD-3DGS degrades, leading to blurry lane markings that pose challenges for end-to-end driving systems that rely on such cues. Furthermore, we include a visual comparison between ReconSimulator and Street Gaussians in Fig. 8.
>
>
> > **[W2] Limited contribution in the RL algorithm.**
>
> [A2] Our main contribution focuses on exploring the paradigm of enhancing reinforcement learning through diffusion-based scene reconstruction. Additionally, we will open-source the entire reconstruction environment, which can significantly provide a valuable resource for training and evaluating end-to-end autonomous driving algorithms. Furthermore, we provide additional details and analysis of the synthetic data training strategy to enhance the training process, available in Appendix D.
>
> **To be continued in next reply.**

---

> ### Author Response · Authors · 2025-11-24
> **Rebuttal by Authors (continued)**
>
> > **[Q1] ...observe any forgetting issues when training on the synthetic data, if tested back into the real domain with nominal driving cases?**
>
> [A3] We present an experiment in which models are trained on synthetic data and evaluated on real-domain nominal driving cases. In the table below, the amount of real data is held constant, and only the quantity of synthetic data is varied to adjust its proportion in the overall dataset. We do not observe any forgetting issues when using a large amount of synthetic data. On the contrary, there is a stable performance improvement on real-domain data in nominal driving scenarios. This consistent enhancement can largely be attributed to the high-quality scene reconstruction achieved by the ReconSimulator. We have added this analysis in the revision (see Appendix D).
>
> | Real Data | Synthetic Data | CR↓   | DR↓   |
> | :--------------------------------: | :-----------: | :-----------: | :---------: |
> | 100%     | 0%             | 0.362 | 0.231 |
> | 80%      | 20%            | 0.276 | 0.214 |
> | 60%      | 40%            | 0.214 | 0.182 |
> | 40%      | 60%            | 0.155 | 0.154 |
> | 20%      | 80%            | **0.123** | **0.130** |
>
>
> > **[Q2] ...What is the function specifically defined in the DAA module? What is the sample scale needed for DAA to get interesting adversaries? If we conduct random samples, would most of the samples remain boring without challenging behaviors?**
>
> [A4] We use a text-to-trajectory model as the function for the DAA module. Specifically, we first construct a comprehensive trajectory-generation function library that covers a wide range of agent behaviors, such as steering, acceleration, and braking. Leveraging this library, we curate a paired text-to-trajectory dataset to fine-tune a large language model (LLM), enabling it to automatically generate diverse agent trajectories from natural language descriptions. We provide more details in Appendix E, including the full pipeline used to generate trajectories and example code for the trajectory-generation function library.
>
> Meanwhile, DAA can efficiently generate interesting adversarial scenarios based on the functions and original trajectories. We experimented to explore the relationship between the number of samples and the number of generated interesting adversarial scenarios:
>
> | Number of Samples | Number of Challenging Scenarios | Percentage of Challenging Scenarios |
> | :--------------------------------: | :-----------: | :-----------: |
> | 1000              | 435                             | 43.5%                               |
> | 2000              | 892                             | 44.6%                               |
> | 4000              | 1653                            | 41.3%                               |
> | 8000              | 3470                            | 43.4%                               |
> | 16000             | 6712                            | 42.0%                               |
>
> Therefore, when performing random sampling, most of the samples will not lack challenging behaviors, with a significant portion being highly challenging. Meanwhile, the remaining scenarios are still important, and discarding them in favor of keeping only the challenging ones (e.g., collision-inducing scenarios) would negatively impact the performance of reinforcement learning.

---

> ### Author Response · Authors · 2025-11-24
> **Rebuttal by Authors (continued)**
>
> > **[Q3] In the online rollout in the PPO, what is the data diversity, sample scale, as well as their impact on the final performance? An ablation study on the final performance vs sample complexity will be interesting.**
>
> [A5] In the table below, we investigate the relationship between final performance and sample complexity.  We measure sample complexity using the scenario count, where each change in surrounding-vehicle trajectories or ego navigation (modified by CTG or DAA) is treated as a new scenario. Our results show that a higher sample complexity generally leads to stronger final performance, as it provides richer training diversity and exposes the policy to a broader range of challenging situations.
>
>
> | Scenario Count | CR↓   | DR↓   |
> | :--------------------------------: | :-----------: | :-----------: |
> | 1000           | 0.101 | 0.063 |
> | 2000           | 0.092 | 0.051 |
> | 4000           | 0.087 | 0.044 |
> | 8000           | 0.081 | 0.032 |
> | 16000          | 0.077 | 0.024 |
> | 32000          | **0.071** | **0.023** |
>
> Additionally, we conducted a further analysis of the DAA module, specifically investigating the impact of the number of extreme case samples on final performance. In this experiment, we kept the number of non-corner-case scenarios constant and only varied the number of corner-case scenarios. The results, shown in the table below, demonstrate that the quantity of corner cases significantly affects the final performance, particularly the collision rate. Increasing the number of corner cases helps reduce the collision rate, but may lead to instability in the DR  metric. This is because a large number of corner cases tends to drive the policy towards more conservative behavior, which, while reducing crashes, may lead to hesitation or failure to accelerate in situations where it is necessary, ultimately resulting in yawing.
>
> | Corner Case Count | CR↓   | DR↓   |
> | :--------------------------------: | :-----------: | :-----------: |
> | 4000           | 0.072 | 0.046 |
> | 8000           | 0.078 | 0.037 |
> | 12000          | 0.065 | 0.034 |
> | 16000          | 0.063 | **0.032** |
> | 20000          | 0.061 | 0.036 |
> | 24000          | 0.059 | 0.037 |
> | 28000          | **0.057** | 0.042 |

---

> ### Author Response · Authors · 2025-11-25
> **Rebuttal by Authors (continued)**
>
> > **[Q4] Are DAA and CTG used only to cold-start the training data for RL, or also incorporated during the RL process? If they are both used during online learning, how are they used?**
>
> [A6] Both DAA and CTG are also incorporated during the RL process. Specifically, in the reinforcement learning training phase, we alternate between reinforcement learning and imitation learning training processes. During the reinforcement learning phase, we first sample data from the collected dataset and apply the Proximal Policy Optimization (PPO) algorithm to update the policy. The PPO algorithm optimizes the policy, enabling the agent to better respond to environmental changes, thereby enhancing the performance of the policy. Meanwhile, during the imitation learning phase, we train the model using data from three primary sources: first, the nuScenes dataset, which provides complex real-world driving scenarios; second, edge-case data generated by DAA, which includes many challenging driving situations; and third, the Cousin-nuScenes dataset, an extension of the nuScenes dataset that provides additional scene samples. These diverse data sources ensure that the model can be effectively trained in various scenarios, maintaining strong generalization capabilities. We also conducted an analysis of the impact of the ratio between reinforcement learning and imitation learning during the reinforcement learning training phase on the final results. We found that when the ratio is 4:1, the collision rate is the lowest. In the revision, we have added the training pipeline in Fig. 9 and substantially refined the description of the *ReconDreamer-RL* training procedure in Appendix D.
>
> | RL:IL | CR↓   | DCR↓  | SCR↓  | DR↓   | PDR↓  | HDR↓  |
> |:-----:|:-----:|:-----:|:-----:|:-----:|:-----:|:-----:|
> | 0:1   | 0.238 | 0.143 | 0.095 | 0.084 | 0.057 | 0.027 |
> | 1:0   | 0.111 | 0.069 | 0.042 | 0.057| 0.037 | 0.020 |
> | 2:1   | 0.091 | 0.056 | 0.035 | 0.055 | 0.034 | 0.021 |
> | 4:1   | **0.077** | **0.048** | **0.029** | **0.040** | **0.027** | **0.013** |
> | 8:1   | 0.090 | 0.058 | 0.032 | 0.042 | 0.025 | 0.017 |
>
>
> ---
>
> We sincerely appreciate Reviewer AWCj for your thorough and valuable feedback. We have carefully considered all of your points and made every effort to address each concern in our responses. If you have any further questions or need additional clarification, please don’t hesitate to reach out. We look forward to continuing the discussion during the rebuttal period and value your further insights!

---

> > ### Comment · Reviewer_AWCj · 2025-11-26
> >
> > I thank the reviewer for their comprehensive response on (1) the design details of ReconSimulator, (2) the impact on data diversity and sample size on the PPO, and (3) further clarification on the DAA and CTG modules.
> >
> > - How does the author define "interesting adversarial scenarios"? The LLM-guided generation has an>40% interesting rate, which seems to be very high given the complex nature of urban driving scenes. Is the definition of "interesting adversarial scenarios" the same as "extreme case samples" in the response to Q3?
> >
> > - In the response to Q3, the authors analyze the PPO's final performance under different scales of samples , and different scales of extreme case samples. The authors fail to explain in which domain the DR and CR are evaluated, e.g., the nominal real driving domain or synthetic/adversarial scenarios? Also, in the second table, how is the corner case defined? Is it defined with respect to the current ego behavior? In the second table, is CTG not applied?
> >
> > I appreciate the authors' effort in their initial response. I would make my final evaluation after the authors clarify the follow-up questions above.

---

> ### Author Response · Authors · 2025-11-28
> **Rebuttal by Authors**
>
> Thank you for your further questions, which have helped us refine the details of our research! Our response is as follows, and we have made additional revisions to the paper. The changes are highlighted in light pink.
>
> ---
> > **[Q1] How does the author define "interesting adversarial scenarios"? The LLM-guided generation has an>40% interesting rate, which seems to be very high given the complex nature of urban driving scenes. Is the definition of "interesting adversarial scenarios" the same as "extreme case samples" in the response to Q3?**
>
> [A1]  To define "interesting adversarial scenarios," we use the frozen-parameter policy trained during the imitation learning phase to determine whether the predictions in these edited scenarios differ from those in the unedited scenarios. The prediction window is set to 0.5 seconds, and we classify a scene as an "interesting adversarial scenario" if the deviation exceeds 0.5 meters in the lateral direction or 1 meter in the longitudinal direction. Additionally, when the lateral and longitudinal distances between the selected target vehicle and the ego vehicle are below the corresponding thresholds, such as 1 meter laterally and 2 meters longitudinally, the scenario is also considered relevant. Furthermore, the high ratio observed can be attributed to the careful selection of target vehicles and the physical constraints within the scene. For example, when identifying potentially interactive target vehicles, such as a vehicle cutting in from the left rear, this significantly narrows down the feasible interactions based on direction. Moreover, the speed and other characteristics of different vehicle types must adhere to physical laws, which further limits the set of vehicles capable of realistically engaging in interactions. With these multiple constraints in place, identifying vehicles that can genuinely interact becomes relatively straightforward.
>
> Yes, this is the same. We have revised the paper to ensure consistency in terminology, and we have also added the definition of "interesting adversarial scenarios" for clarity.
>
>
> > **[Q2] In the response to Q3, the authors analyze the PPO's final performance under different scales of samples , and different scales of extreme case samples. The authors fail to explain in which domain the DR and CR are evaluated, e.g., the nominal real driving domain or synthetic/adversarial scenarios? Also, in the second table, how is the corner case defined? Is it defined with respect to the current ego behavior? In the second table, is CTG not applied?**
>
> [A2] Thank you very much for your suggestion to add more details in response to Q3. The experiments on PPO’s final performance, conducted across different sample scales and extreme-case samples in the A5, are performed in the real driving domain. Meanwhile, we provide the results for the corner cases in Tables 3 and 4. In the second table in A5, corner cases are defined based on the current ego behaviour. Additionally, CTG is not applied in this table to isolate its effect and validate the impact of corner cases on the final results. These details have been added to our revised paper. Once again, thank you for helping us refine our work!
>
>
> **Table 3: Relationship between scenario count and final performance in synthetic adversarial scenarios.**
> | Scenario Count | CR↓   |
> | :--------------------------------: | :-----------: |
> | 1000           | 0.172 |
> | 2000           | 0.154
> | 4000           | 0.131 |
> | 8000           | 0.107 |
> | 16000          | 0.096 |
> | 32000          | **0.081** |
>
> **Table 4: Relationship between interesting adversarial scenarios count and final performance in synthetic adversarial scenarios.**
> |  Interesting Adversarial Scenarios | CR↓   |
> | :--------------------------------: | :--------------------------------: |
> | 4000           | 0.132 |
> | 8000           | 0.111 |
> | 12000          | 0.097 |
> | 16000          | 0.083 |
> | 20000          | 0.079 |
> | 24000      | **0.072** |
> | 28000          | 0.073 |
>
> ---
>
> We would like to express our sincere appreciation for your thorough and thoughtful review, as well as your valuable feedback, which has greatly helped us refine the details of our paper！

---

### Author Response · Authors · 2025-11-24
**Overall Author Rebuttal**

We sincerely thank the reviewers and chairs for their time and thoughtful feedback. We greatly appreciate the recognition of the originality, motivation, technical contributions, solid empirical results, and scalability of our work. In response to the reviewers' suggestions, we have revised the paper, with the changes highlighted in light green background for clarity.

---

### Author Response · Authors · 2025-12-02
**Final Summary by Authors**

Dear PCs, SACs, ACs,

We are deeply grateful to all participants, whose efforts have greatly enhanced our work.

We are happy that all reviewers recognize the motivation, the problem formulation, the strong empirical results, and the scalability as convincing. Our work introduces a novel paradigm for addressing the sim2real gap, with a particular focus on the scarcity of corner cases. We begin by integrating video diffusion priors with a kinematic model to reconstruct more realistic driving environments, demonstrating that this approach is cost-effective and significantly enhances simulation realism. Additionally, we propose an agent for automatic corner case generation and train an end-to-end policy in the reconstructed environment using reinforcement learning. As a result, we achieve a five-fold reduction in the Collision Ratio. Our approach is thoroughly validated through extensive ablation studies, scalability assessments, and generalization experiments.

We have also addressed all concerns. In response to the shared request for implementation details of the Dynamic Adversary Agent (DAA) module, we have added complete specifications in the appendix, including the text-to-trajectory function, BEV-space collision and out-of-bounds checks, pipeline figures, and pseudocode.

Reviewer AWCj suggested adding technical details of the ReconSimulator and recommended a systematic analysis of corner-case prevalence and the impact of the synthetic-to-real data ratio on final performance. In Appendices C and D, we have now provided a two-stage reconstruction pipeline, integrated a diffusion prior, and conducted multiple ablation experiments that directly address these points, as acknowledged by the reviewer: "Appreciate the authors’ effort in their initial response." The reviewer also promised to make their final evaluation after we clarify the experiment settings, which have been explained in detail.

Following Reviewer mSxh’s suggestion, we have added closed-loop simulations beyond our in-house reconstruction environment (e.g., nuPlan) and incorporated driving-quality metrics, such as longitudinal and lateral jerk. The additional results have been included in the main text.

Regarding Reviewer d1uV’s concern about the computational overhead of iterative reconstruction in ReconSimulator, Appendix C reports extra optimization steps and per-scene reconstruction time under different training settings, showing that the diffusion prior introduces only a lightweight, acceptable overhead. This reviewer also noted the sufficiency and clarity of our response in the discussion.

We have addressed all reviewers’ concerns with detailed explanations and have made the required revisions in the main text. We believe our work makes a significant advancement to the field and will release the code, data, and checkpoints to support further research.

We sincerely thank you for your time and careful consideration.

Best regards,

The Authors

---

### Meta-Review · Area_Chair_Vrdr · 2026-01-06

**Summary:**

This paper proposes "ReconDreamer-RL," a framework enhancing end-to-end autonomous driving training by combining a 3DGS-based simulator with video diffusion priors and kinematic modeling. The authors also introduce a Dynamic Adversary Agent (DAA) and a Cousin Trajectory Generator (CTG) to enrich training data with corner cases and diverse trajectories.

While the engineering effort to integrate these components is substantial and the empirical results show improvements in collision rates within the simulated environment, the consensus among the committee—particularly regarding the novelty and the robustness of the evaluation definitions—prevents me from recommending acceptance. The primary concerns informing this rejection are the incremental nature of the technical contribution (stitching together existing methods like DriveDreamer4D and standard PPO) and outstanding ambiguity regarding the definitions of adversarial scenarios used to claim performance gains. Despite a thorough rebuttal, the core concern regarding whether the "interesting" scenarios are truly challenging or simply artifacts of the generation process remains unresolved.

**Reviewer Concerns:**

**Addressed Concerns**

(1) The authors successfully clarified the technical specifications of the ReconSimulator and the two-stage training pipeline (Imitation Learning followed by RL).

(2) The request for details on the text-to-trajectory implementation and collision checking in the Dynamic Adversary Agent was largely met with the inclusion of Appendix E and pseudocode.

(3) The authors addressed Reviewer mSxh's request for driving quality metrics by adding jerk analysis and expanded the evaluation to include NuPlan closed-loop experiments.

(4) Reviewer d1uV’s concern regarding the overhead of the iterative reconstruction was addressed with runtime comparisons.

**Outstanding Concerns**

(1) Technical Novelty (Reviewer BUDD & AWCj): This is the most critical outstanding issue. As noted by Reviewer BUDD, the individual components (diffusion for reconstruction, decoupled scene representation, adversarial agents) have been explored in prior works (e.g., DriveDreamer4D, OmniRe, DiffScene). The combination, while useful, is viewed as an engineering integration rather than a fundamental algorithmic advance. The RL component relies on standard PPO, which limits the learning-theoretic contribution.

(2) Definition of "Adversarial" (Reviewer AWCj): In the final discussion phase, Reviewer AWCj rightly questioned the definition of "interesting adversarial scenarios." The authors defined this based on a deviation threshold (>0.5m lateral) and reported a >40% hit rate for generated scenarios being "interesting." I agree with the reviewer that this rate seems suspiciously high for complex urban driving, suggesting the definition of "interesting" might be too loose or heuristically tuned to favor the proposed method. This casts doubt on the difficulty of the evaluation benchmarks.

Sim-to-Real Gap: While the paper claims to bridge the sim-to-real gap, the evaluation remains heavily dependent on the authors' own reconstructed environment. The circularity of training and testing on distributions generated by the same pipeline (even with NuPlan added as a validator) remains a validity threat that was not fully dispelled.

**Reviewer Scores:**

The reviews were mixed, and while some scores were technically above the threshold, the substance of the critique points toward rejection. Here is my assessment of where the scores would land given the final exchanges:

Reviewer AWCj (Score: 4): This reviewer remained the most critical regarding the technical depth and definitions. They acknowledged the clarifications but explicitly raised new, fundamental doubts about the "interesting adversarial scenarios" definition in their final comment. I think they would maintain their score of 4.

Reviewer BUDD (Score: 6 -> 6/4): This reviewer flagged the lack of originality as a primary weakness. While they gave a "marginally above" score initially, the rebuttal did not fundamentally change the fact that the method is an assemblage of existing techniques. I estimate they would lower the score upon closer scrutiny of the novelty compared to the cited works.

Reviewer mSxh (Score: 6): This reviewer was generally positive about the empirical gains and the addition of NuPlan results. They would likely maintain a 6.

Reviewer d1uV (Score: 6): This reviewer’s concerns were mostly regarding details and computational costs, which were answered. They would likely maintain a 6.

---

### Decision · Program_Chairs · 2026-01-26

Reject